# A Review of the Complex Flow and Heat Transfer Characteristics in Microchannels

**DOI:** 10.3390/mi14071451

**Published:** 2023-07-19

**Authors:** Junqiang Zhang, Zhengping Zou, Chao Fu

**Affiliations:** 1School of Energy and Power Engineering, Beihang University, Beijing 100191, China; zhangjq6775@163.com; 2National Key Laboratory of Science and Technology on Aero-Engine and Aero-Thermodynamics, Beihang University, Beijing 100191, China; 3Research Institute of Aero-Engine, Beihang University, Beijing 102206, China

**Keywords:** microchannels, single-phase, scaling effects, phase change, flow boiling, pseudophase change, flow structures, heat transfer mechanism

## Abstract

Continuously improving heat transfer efficiency is one of the important goals in the field of energy. Compact heat exchangers characterized by microscale flow and heat transfer have successfully provided solutions for this purpose. However, as the characteristic scale of the channels decreases, the flow and heat transfer characteristics may differ from those at the conventional scale. When considering the influence of scale effects and changes in special fluid properties, the flow and heat transfer process becomes more complex. The conclusions of the relevant studies have not been unified, and there are even disagreements on some aspects. Therefore, further research is needed to obtain a sufficient understanding of flow structure and heat transfer mechanisms in microchannels. This article systematically reviews the research about microscale flow and heat transfer, focusing on the flow and heat transfer mechanisms in microchannels, which is elaborated in the following two perspectives: one is the microscale single-phase flow and heat transfer that only considers the influence of scale effects, the other is the special heat transfer phenomena brought about by the coupling of microscale flow with special fluids (fluid with phase change (pseudophase change)). The microscale flow and heat transfer mechanisms under the influence of multiple factors, including scale effects (such as rarefaction, surface roughness, axial heat conduction, and compressibility) and special fluids, are investigated, which can meet the specific needs for the design of various microscale heat exchangers.

## 1. Introduction

The escalating energy demand has led to a gradual increase in resource scarcity and environmental degradation. Consequently, policymakers and regulators in many countries have opted to prioritize the enhancement of energy conversion efficiency and the restructuring of energy distribution. This has resulted in a surge of research interest in improving the efficiency of heat engines and heat exchangers [1]. In particular, compact heat exchangers are gradually replacing traditional heat exchangers, which can ensure efficient heat exchange tasks under more stringent space and weight constraints. Compact heat exchangers are usually composed of micro/small heat exchange units or microscale channels. Compared to conventional large-scale channels, they have an extremely high surface area-to-volume ratio and several orders of magnitude higher heat transfer performance [2]. Compact heat exchangers have become the preferred choice in many practical applications. With the improvement of manufacturing technology and heat transfer enhancement technology, various new types of microchannel heat exchanger configurations have been proposed and widely applied in many fields, as shown in Figure 1, including nuclear energy [3,4], solar thermal [5], molten salt fuel cells [6,7], microelectronic micro-mechanical systems [8,9], automotive cooling systems [10,11], aerospace (active regenerative cooling systems, precooled propulsion systems, etc.) [12,13,14,15,16,17,18,19,20,21], and so on.

The working state of compact heat exchangers can be attributed to the physical process of microscale flow and heat transfer. Research on microscale flow can be traced back to the 1840s, when Poiseuille et al. [22] published their first paper describing flow in pipes with diameters ranging from 30 to 150 mm. In 1909, Knudsen et al. [23] studied gas flow through glass capillaries in transition and free molecular flow schemes. In 1913, Gaede et al. [24] conducted the first known parallel microchannel flow experiment. Microfluid devices have a significant advantage in heat transfer due to their extremely high surface area-to-volume ratio, which naturally attracts the attention of researchers in the field of heat transfer. In 1981, Tuckerman et al. [25] first proposed the concept of microchannel heat sinks and demonstrated that reducing channel size can significantly improve the heat transfer performance of heat sinks. Afterward, with the urgent demand for compactness in different industrial systems, by the 1990s, microchannel fluid dynamics received widespread attention and created a scientific research field for microfluidics [26].

Despite the increasing applications of microscale heat transfer in science and engineering, including constant property fluid type, boiling type, supercritical fluid type, nano-fluid type, etc., and the extensive research that has been conducted in this field, no unified conclusion has been reached on the fluid dynamics and heat transfer processes. Therefore, it is necessary to deeply understand the characteristics of microscale flow and heat transfer and explore the mechanism of microscale flow and heat transfer under the influence of multiple factors including scale effects (rarefaction effects, surface roughness, axial heat conduction, compressibility effects, etc.) and special fluids, which will meet the various requirements for the design of different microscale heat exchangers.

The literature on microchannel heat exchangers can be classified based on fluid flow characteristics, heat transfer mechanisms, boundary condition types, and solution methods [27], as shown in Figure 2. This article focuses on the mechanism of flow and heat transfer in microchannels, so it will be elaborated from the following two perspectives. The first type is the microchannel flow and heat transfer mechanism that only considers the influence of scale effects in the single-phase fluid flow through microchannels. The second type is the complex flow and heat transfer mechanism reflected by the coupling of microchannel flow and special heat transfer phenomena brought about by special fluids (fluid with phase change (pseudophase change)). A systematic review was conducted on the current research status of microchannel flow and heat transfer, and the results were summarized.

## 2. Flow and Heat Transfer Mechanism of Single-Phase Fluid under Scale Effects

### 2.1. Criteria for Microchannels

The definition of channel size is of great significance for studying flow and heat transfer characteristics at different scales, and different scholars have different definitions of channel sizes. At present, the commonly used criteria for defining channel scales are Kandlikar and Grande [28]. The threshold and scale definitions are detailed in Table 1. The threshold is determined to be a hydraulic diameter of less than 3 mm. This article also uses these criteria to distinguish between macrochannels and microchannels. In addition, some scholars divide macro- and microchannels through several important dimensionless numbers, such as Bo number [29], Co number [30], and Eo number [31].

### 2.2. Impact Mechanism of Scale Effects

Generally speaking, the forces controlling flow and heat transfer processes are closely related to the characteristic scale, and the relationship between different forces and scales is shown in Table 2. In the process of microscale flow and heat transfer, as the characteristic scale decreases, compactness increases, and the influence of area-related forces (such as surface tension, viscous forces, etc.) relatively increases, while the influence of volume-related forces (such as buoyancy, gravity, etc.) relatively weakens, presenting a flow and heat transfer law different from that of conventional scales [32].

The reason for the difference in flow and heat transfer characteristics from conventional scales caused by the reduction of system scale can be attributed to scale effects. The factors that usually have a small effect or are ignored in conventional scale channels, such as axial heat conduction, surface roughness, compressibility effects, viscous heating, and entrance effects, etc., may have a significant impact on flow and heat transfer in microchannels. Therefore, scale effects need to be given special attention. The physical mechanisms of scale effects can be divided into two categories [32]: the first type is that when the scale of the system is equal to or greater than the average free path of molecules, the assumption of a continuous medium at conventional scales is no longer valid, and a new basic formula describing flow and heat transfer laws needs to be modified or established. The second type is that the basic equations and laws at all conventional scales are still applicable, but due to the reduction of scale, the relative importance of various factors affecting flow and heat transfer has changed, resulting in different phenomena and laws. In addition, early research on microchannel flow and heat transfer was mainly focused on experiments. Due to the limited experimental measurement methods and technical level at that time, the measurement uncertainty of some experiments may be high, leading to deviations from traditional theories such as friction coefficients and heat transfer correlations in microchannels. However, with the continuous improvement of processing technology and the emergence of high-precision measurement methods, some microchannel experiments that were previously believed to deviate from traditional theoretical predictions were analyzed again using high-precision equipment. It was found that the friction coefficient obtained in the new experimental data was very consistent with the traditional value (fRe = 64) [33,34]. However, measurement uncertainties cannot always explain all reported discrepancies. When the length scale is reduced to the micron range, some physical phenomena that are usually neglected in conventional channels may have a significant impact on microscale flow.

A summary of the relevant research on scale effects in microchannels is presented in Table 3. Some of them need to be explained in detail later. The scale effects of microchannels are examined in detail below, with a review of recently published results.

#### 2.2.1. Rarefaction Effects

The rarefaction effects are the phenomenon where classical continuity theory, no-slip, and no-temperature jump conditions are no longer strictly applicable when the mean free path (λ) of fluid molecules is comparable to the characteristic length of the system (L). Some molecules that reflect from the wall penetrate without fully interacting with the layer of gas molecules near the wall and the wall information is not fully transmitted in the adjacent gas layers. The result is that the velocity and temperature of the adjacent fluid differ from the physical quantities at the wall, manifesting as velocity slip and temperature jumps, as shown in Figure 3 [51]. The slip velocity and temperature jump boundary conditions of microchannels can be defined as follows [52]:(1)$$u_{s}=-\frac{2-\sigma_{m}}{\sigma_{m}}\lambda {\left(\frac{du}{dr}\right)}_{r=R}$$
(2)$$T-T_{s}=-\frac{2-\sigma_{t}}{\sigma_{t}}\frac{2\gamma }{\gamma +1}\frac{\lambda }{\mathrm{Pr}}{\left(\frac{\partial T}{\partial r}\right)}_{r=R}$$
where $\sigma_{m}$ is the tangential momentum accommodation coefficient, $\sigma_{t}$ is the thermal accommodation coefficient, and γ is the specific heat ratio. The above two formulas take into account the discontinuous effects caused by rarefaction, and most existing studies on microchannel heat transfer have successfully used these equations.

The degree of rarefaction is usually distinguished by the Knudsen number (*Kn*), which is defined as the ratio between the mean free path of molecules and the characteristic length scale [53].
(3)$$\mathrm{Kn}=\frac{\lambda }{L}$$

For example, considering the airflow in a centimeter-sized channel, the *Kn* is $7\times 10^{-6}$ because the average free path of air at standard atmospheric pressure is about 70 nm. Therefore, at the conventional scale, due to the large *L* value, *Kn* → 0, while at the microchannel flow, it can easily be verified that *Kn* is not negligible (>10^−3^).

According to the Knudsen number, the flow can be divided into four states, as shown in Table 4 [54]. The flow is considered to be a continuum flow for small values of *Kn* (<0.001), and the well-known Navier–Stokes equations together with the no-slip and no-temperature jump boundary condition are applicable for the flow field. For 0.001 < *Kn* < 0.1, the flow is in a slip flow regime (slightly rarefied). For 0.1 < *Kn* < 10, the flow is in a transition regime (moderately rarefied). Finally, the flow is considered as a free-molecular flow for large values of *Kn* (>10) (highly rarefied), and tools for solving this type of flow often use molecular dynamics [55,56,57] or Direct Simulation Monte Carlo (DSMC) [58].

Some researchers have studied the effect of rarefaction on microchannel flow and heat transfer through the relationship between *Kn* and *Nu*. Murat et al. [52] demonstrated that the Nusselt number decreases with increasing Knudsen number, which was attributed to the increased wall temperature jump. For low Peclet number values, temperature gradients and temperature jump make the Knudsen number have a large effect on the flow, as shown in Figure 4. The reason for this phenomenon can be attributed to an increase in temperature jump at the wall [35].

In a laminar forced convection study, Sadeghi and Saidi [59] investigated that the effect of viscous heating on the Nu number becomes negligible when the *Kn* number is large, and increasing the *Kn* value without viscous heating leads to a decrease in the *Nu* number. In addition, they observed that viscous heating can cause a singularity in the *Nu* value. Miyamoto et al. [60] and Sheela Francisca [61] also discussed these singularities.

In flows where the rarefaction effects are obvious, velocity slip can cause a decrease in friction coefficient, while temperature jump often leads to a decrease in heat transfer performance. Kavehpour et al. [62] simulated compressibility and rarefaction effects in parallel plate microchannels. The results showed that both *Nu* and *Cf* are much smaller than those of continuous flow under the conditions of uniform wall temperature and uniform wall heat flux. Hettiarachchi et al. [63] investigated the influence of rarefaction effects on laminar flow and heat transfer characteristics in microchannels. They found that velocity slip increases advection near the wall, leading to an increase in heat transfer, while temperature jump increases thermal resistance, leading to a decrease in heat transfer. The combined effect may lead to an increase or decrease in Nu numbers, depending on their relative size. The influence of relevant parameters on the local Nusselt number distribution is shown in Figure 5. Ejtehadi [64] analyzed the influence of compressibility and rarefaction on micro/nanoscale Couette flow from the perspective of entropy generation. Research showed that the increase of Mach number leads to non-uniform entropy profiles near the central region of the channel. In addition, entropy production had significantly increased in all regions of the domain stages. By contrast, increasing the Knudsen number would have the opposite effect. Tang et al. [36] found that the friction coefficient of fused silica microtubes (D = 10–20  μm) is lower than the theoretically predicted value through experiments, and the decrease in friction coefficient is believed to be caused by the rarefaction effect.

In the prediction of microchannel gas flow and rarefaction effects, Nishanth et al. [65] proposed a new continuous medium solution method: the extended Navier–Stokes–Fourier equations. This approach was able to capture the physical details of microscale gas flows over a wide range of Knudsen numbers without using any slip/jump boundary conditions. Van Rij, J. et al. [66] studied the convective heat transfer process in rectangular microchannels based on the assumption of continuum and applied the slip velocity and temperature jump boundary conditions to the momentum equation and the energy equation, respectively. The performance of the standard lattice Boltzmann method (LBM) was confined to the microchannel flows with a Knudsen number lower than 0.1. Isfahani et al. [67] proposed an improved LBM method, which can realize the calculation of the Knudsen number flow in a larger range. Research has shown that micro/nanochannels filled with porous media exhibit the minimum Nussen number effect. In addition to porosity and Knudsen number, the size of obstacles also played an important role in heat transfer. Therefore, when the size of obstacles decreased, heat transfer enhancement could be observed.

#### 2.2.2. Surface Roughness

The relative surface roughness can be defined as the ratio between the roughness of the surface material (ε) and the hydraulic diameter (d_h_) of the microchannels, as shown in Figure 6. In conventional scale channels, the importance of surface roughness is only significant in turbulent regions. When the wall roughness is less than 5% of the channel diameter, its impact on laminar flow in conventional channels can be ignored. However, for microchannels, even in the laminar flow zone, surface roughness may have a significant impact.

The effects of surface roughness on pressure drop have been widely studied in the literature. Moody described Darcy’s friction factor f_Darcy_ as a function of Reynolds number and relative roughness and completed the Moody diagram. Although the relative roughness values currently used in the Moody diagram are only up to 5%, in many applications, such as high heat flux cooling and microflow, the flow in small-diameter channels is expected to have higher values. Compared to large channels, fluid flow in microchannels may differ in terms of wall friction effects. The base diameter D_t_ of the channel is generally used to calculate the Moody diagram. Taking Nikuradse’s experiment as an example, the base diameter D_t_ was used in their pressure drop calculation [68]. Although this error is relatively small for the range of roughness used in the Moody diagram, it can become significant if the diameter correction is not taken into account when extending the Moody diagram for the channels with high relative roughness values. Therefore, Kandlikar et al. [69] used the constriction diameter to replace the base diameter in both laminar and turbulent regions, so as to reflect the constriction effects caused by the relative roughness in the microchannels, and then obtained the modified Moody diagram suitable for the microchannel, as shown in Figure 7.

In terms of experimental studies, Mala et al. [38] conducted experiments on stainless steel and fused silica microtubules with diameters of 50–254 μm and a relative roughness of 0.69–3.5%. The Reynolds number range of the test was 100 to 2000. It was found that for smooth microtubules with small diameters, the flow characteristics are significantly different from those predicted by traditional theories. The pressure drop when the Reynolds number is less than 1000 is basically consistent with the pressure drop predicted by Poiseuille flow theory. When the Reynolds number is greater than 1000, the pressure gradient increases significantly compared with that predicted by Poisouille flow theory. Liu et al. [70] found that surface roughness has a significant impact on the flow and heat transfer performance of microchannels. The microchannel feature scale was 0.4 mm, with a relative roughness of 0.58% to 1.26%. Based on the experimental data, the empirical correlations of flow resistance and heat transfer characteristics in microchannels were established, respectively. Wagner et al. [71] established a theoretical model to predict the effects of roughness pitch and height on the pressure drop along the channel flow direction. Zhou et al. [72] found that all the normalized data of fRe can be predicted by the original constricted flow model within an error of about 15%. Lin et al. [73] designed four different surface roughness features. The results indicated that roughness has little effect on the flow and heat transfer characteristics in laminar flow. However, in turbulent conditions, heat transfer was significantly enhanced due to the roughness effect. Dai et al. [74] established a database based on the experimental results of liquid flow in rough microchannels, including 5569 data, covering a wide range of relative roughness and Reynolds numbers. A universal threshold had been proposed to subdivide the flow into smooth and rough microchannels, and new correlations for predicting friction coefficient and critical Reynolds numbers were obtained. Thiago et al. [75] investigated the effect of surface roughness on the mass flow rate inside adiabatic capillaries through experiments. Mandev et al. [42] experimentally studied the influence of surface roughness induced by the manufacturing process on mixed convective heat transfer in microchannels (300–700 μm). The surface maps taken from the optical profilometer are shown in Figure 8a, and the examples for microchannel cross-sections taken from the optical microscope are presented in Figure 8b. The results showed that the effect of surface roughness on forced convection heat transfer is more significant than that of natural convection heat transfer. This effect became apparent as the hydraulic diameter decreased. The increase of heat transfer by surface roughness could be attributed to the increase in surface area, mixing effect, and boundary layer interaction.

To better elucidate the influence of roughness on fluid flow and heat transfer characteristics, describing surface defects through numerical simulation is an effective method that can isolate the roughness effect from other factors formed in microchannels. [41,42,76,77]. For establishing numerical methods, accurately describing the random roughness characteristics of material surfaces is the most critical aspect of building physical models. The primary challenge is how to accurately describe the complex geometry of the random roughness present in microchannels. In microchannels, roughness typically appears in a random fashion, with the shape, size, and position of roughness elements being unpredictable. The Gaussian function has been utilized to solve these challenges. As of now, various roughness models have been proposed, including 2D and 3D models. Kleinstreuer et al. [78,79] proposed a computational model named porous medium layer (PML) to study the influence of surface relative roughness on the friction effect of laminar flow in microchannels. Surface roughness was described as a random distribution of peak and valley values of wall distance, simulated as a uniform porous medium layer characterized by the porosity α and the height h, as shown in Figure 9. The fluid variables were evaluated as a function of PML characteristics, and the model took into account the realistic values of PML Darcy number, relative surface roughness, and flow area so as to match the friction factors observed in microchannels. For systems with significant relative roughness values, the predicted results of the model were in good agreement with the measured dataset. Croce et al. [80] proposed a model for roughness by generating a collection of peak valley values randomly on the surface of a smooth tube. A three-dimensional Gaussian roughness model with high efficiency and accuracy was proposed by Guo et al. [81]. Results demonstrated that both flow resistance and heat transfer are sensitive to the surface morphology of the microchannels.

In general, roughness plays a positive role in thermal performance as well as flow resistance under laminar flow. A general numerical method for creating random irregular roughness based on the roughness profile was developed, and the Gaussian function was utilized for the roughness profile [82]. Four rough mesh surfaces generated with an auto correlation length (ACL) are shown in Figure 10. The effect of surface roughness with several heights on the flow and heat transfer characteristics was investigated at different Reynolds numbers. The research showed that changes in the viscous shear stress and the heat transfer coefficient follow the roughness structure, increasing over roughness peaks and decreasing in roughness valleys, as shown in Figure 11 and Figure 12. In terms of roughness modeling, Ansari et al. [83], Li et al. [84], Gamrat et al. [85], and Croce et al. [86,87] proposed different methods, respectively. Majumdar et al. [88] used fractal to characterize the multiscale self-affine topography by scale-independent parameters such as the fractal dimension. In the subsequent studies, Chen et al. [89] introduced the Weierstrass Mandelbrot function to simulate fractal surfaces. Their findings indicated that pressure drop tends to increase as the relative roughness and fractal dimension increase, which is attributed to the flow recirculation and separation that occurs due to the presence of rough surfaces. The influence mechanism of flow recirculation and separation on pressure drop in laminar channels can be summarized as follows [90]: when the relative roughness height is small (k/D = 0.01), as shown in Figure 13a, there is no obvious recirculation region anywhere and the friction coefficient is approximately equal to that in the smooth wall case. When the roughness height increases (the local Reynolds number concurrently increases), a clockwise rotating recirculation zone appears near the back-facing wall of the roughness element, as shown in Figure 13b. With increasing roughness height, the size of this recirculation zone increases, and then a stagnation point appears on the front-facing wall of the element, depending on the spacing, which leads to higher pressure losses and increased drag, as shown in Figure 13c. When k/D = 0.1, as shown in Figure 13d, the recirculation zone occupies nearly the entire space between roughness elements and there is very little penetration of the external flow into the cavity. The flow was always steady. In this circumstance, the pressure losses are approximately equal to those in a smooth channel with the same constricted diameter and maximum velocity, which further helps to explain the relative success of using the constricted diameter in the definition of the friction factor.

In addition, accurately solving the governing equations of complex geometric structures formed by random roughness presents a significant challenge. Traditional partial differential equation solvers, such as the finite difference method or finite element method, often become unstable when used to solve these equations. This limits the application of the Navier–Stokes equations to simulate the flows with surface roughness, which can be overcome by using the lattice Boltzmann method (LBM) or Direct Simulation Monte Carlo (DSMC). Pelevic et al. [91] used the lattice Boltzmann method for the first time to study the effect of three-dimensional surface roughness on fluid flow and heat transfer characteristics. It was discovered that the relative roughness has a negligible impact on fluid flow and heat transfer characteristics. When the relative roughness was increased to 2.93%, there was a 7% increase in the Poiseuille number and a 4% increase in the Nusselt number compared to smooth channels. Therefore, the authors did not recommend using surface roughness within microchannels to achieve heat transfer enhancement when there was only a 4% enhancement in heat transfer. Zhang et al. [92] used the lattice Boltzmann method to simulate gas slip flow in rough microchannels and obtained a larger Poiseuille number than that in smooth microchannels. Deng et al. [93] also made the same observation. Additionally, the authors reported that the gas flow characteristics in transition flow regimes are more susceptible to changes in roughness height compared to those in slip flow regimes. The DSMC method can be used to analyze the influence of roughness on flow characteristics in microchannels, as detailed in the relevant studies of Wu et al. [94], Sun et al. [95], and Cao et al. [96].

#### 2.2.3. Axial Heat Conduction

Axial heat conduction is the process of heat transfer within a solid region in the opposite direction to the flow, as shown in Figure 14. Researchers have found that in some heat transfer cases of microchannels, axial heat conduction must be taken seriously [97]. Lin et al. [43] proposed a new parameter to evaluate the impact of axial heat conduction effects in walls. A comparative analysis of literature data [48,98,99] using this model indicated that the axial heat conduction effects of the wall surface play a significant role in gas flow for any channel materials. Cole et al. [100] found that the effect of axial heat conduction in the channel wall is significant under the following conditions: (i) when the microchannel has a small length-over-height ratio; (ii) when the Peclet (*Pe*) number is small; (iii) when the wall thickness relative to the channel height is large; and (iv) when the wall conductivity of the wall material is high relative to the thermal conductivity of the fluid. Yu et al. [101] also found that axial heat conduction dominated the heat transfer process in low *Pe* number flows. With the increase of the *Pe* number, the influence of axial heat conduction gradually decreased until the convective heat transfer was completely dominated.

For the uniformly heated single-phase heat transfer, the wall temperature distribution shows a nonlinear change and deviates from the linear change rule of the ideal condition when the axial heat conduction has a significant impact. If axial heat conduction dominates, the nonlinear distribution of solid and fluid temperatures in the microchannel will become more pronounced [102]. Applying a uniform heat flux to the microchannel does not guarantee a uniform transfer of heat to the fluid. Moreover, the strength of axial heat conduction is directly proportional to the cross-sectional area of the solid domain perpendicular to the direction of flow. An increase in the cross-sectional area of the solid domain will lead to an enhancement of axial thermal conductivity, as shown in Figure 15. Lelea [103] and Moharana [104] both found such nonlinear wall temperature distribution in numerical and experimental studies. This nonlinear wall temperature distribution was also attributed to axial heat conduction. Rahimi [105] discovered that axial heat conduction in the tube wall reduces the local Nusselt number in the entrance and end regions of the microchannel. Three correlations were proposed to predict the ending length and corresponding local and average Nusselt numbers.

In addition to studying the influence of axial heat conduction as a single factor on the flow and heat transfer characteristics in microchannels, some researchers have analyzed the flow and heat transfer process in microchannels under the coupling effect of multiple scale effects. Transient conjugate heat transfer of a hydrodynamically developed and thermally developing laminar flow in a circular microchannel is investigated by Sen et al. [106], considering the effects of rarefaction, viscous dissipation, and axial conduction. The results showed that the *Pe* number, *Kn* number, Brinkman number, and wall thickness ratio all have significant effects on heat transfer characteristics. In microchannel flows, the rarefaction effects, viscous dissipation, and axial heat conduction of the fluid should not be ignored. Aziz et al. [107] studied similar problems of microchannel flows with first-order and second-order slip flow boundary conditions, taking into account axial heat conduction and viscous dissipation. Cetin et al. [37] investigated the Graetz problem in microtubules under slip flow conditions and constant surface temperature boundary conditions by considering the effects of rarefaction, viscous dissipation, and axial heat conduction. Fully developed *Nu* numbers and the thermal entrance length were found to increase with the presence of finite axial conduction. Cetin [108] and Haddout [109] studied the extended Graetz problem from three microscale effects: rarefaction effects, axial heat conduction, and viscous dissipation effects. In addition, there is still a knowledge gap in the appropriate coupling research for all microchannel effects of micro-slit convection. Kalyoncu et al. [110] solved the mathematical challenge by studying the flow heat transfer behavior in micro-slits, considering the complete coupling of rarefaction, axial conduction, and viscous heating. A strong variation of *Nu* in thermal development length was found to dominate the heat transfer behavior of micro-slits with short heating lengths for an early slip flow regime. For this instance, the influence of axial conduction and viscous dissipation was equally important. On the other hand, high *Kn* slip flow suppressed the axial conduction, while viscous heating in a small surface–gas temperature difference case mostly determined the fully developed *Nu* and average heat transfer behavior as a function of *Kn* value.

#### 2.2.4. Compressibility Effects

In microchannel flows where the flow Reynolds number at the inlet is constant, decreasing the characteristic scale leads to an increase in the average fluid velocity. This effect is particularly noticeable in gas flows, where velocity can increase dramatically. As a result of this effect, even when laminar flow conditions are present, compressibility effects can influence the pressure drop. However, in traditional channels, the importance of these effects is usually limited to turbulent conditions (large Reynolds numbers) [111].

It has been noted that decreasing the internal size of a channel may introduce scale effects that must be considered to ensure an accurate evaluation of pressure drop and flow resistance characteristics [112]. Asako et al. [113] investigated the influence of compressibility on gas flow characteristics in two-dimensional microscale channels, and the study showed that the influence of compressibility was significant. It was found that it is mainly a function of Mach number and that it is different from the incompressible value of 96 for parallel-plate channels. Hsieh et al. [114] conducted experimental and theoretical research on the flow characteristics of low Reynolds number compressible gas in microchannels. The microchannel had a characteristic scale of 200 × 50 μm, the Reynolds number ranged between 2.6 and 89.4, and the value of the Knudsen number ranged from 0.001 to 0.02. Their study confirmed the influence of compressibility and rarefaction, with compressibility being dominant. Furthermore, they found that the friction coefficient was lower than that predicted by traditional theory. Chen et al. [115] simulated a three-dimensional steady compressible flow in long microchannels. The results demonstrated that the Knudsen number, compressibility, and aspect ratio all play important roles in predicting the friction characteristics of a microchannel flow. Turner et al. [116] investigated the impact of compressibility on the friction coefficient of air flowing through microchannels. The experiment effectively isolated the effects of surface roughness and rarefaction. The study found that the friction coefficient is consistent with the theory of incompressible continuous flow in extreme cases of low Mach numbers and low Knudsen numbers. However, as the Mach number approached 0.35, the influence of compressibility slightly increased the friction coefficient by 8%. In microtubules, even if the average Mach number is very low, the compressibility effect related to the axial change of gas density becomes very important at large Reynolds numbers and small diameters [117]. High pressure drop and high aspect ratio generate significant compressibility effects in microchannel flow, which compete with the rarefaction effect and relative roughness at small scales [118]. Duan et al. [119] investigated the effect of compressibility on slip flow in non-circular microchannels and proposed a simple model to predict the pressure distribution and mass flow rate of slip flow in most non-circular microchannels. Morini et al. [120] studied the influence of compressibility on the average friction coefficient of nitrogen flow in microtubules ($D_{h}=100-300\;\mu m$). The average friction coefficient under the assumption of isothermal flow was obtained and compared with the f⋅Re correlation proposed by Asako et al. [113].

Due to the high frictional head loss in microchannels, gas density may change significantly between the inlet and outlet, resulting in changes in velocity distribution. Compressibility effects on the friction factor of gaseous laminar flows in a microtube partly result from changes in the velocity profile that must occur to maintain the mass velocity profile when density changes in the radial direction. The remaining compressibility effect can be attributed to actual mass transfer in the radial direction [121]. Celata et al. [122] showed in their study of compressible flow in microchannels that the effect of acceleration caused by density changes is extremely limited, and the assumption of incompressible flow is valid under the flow conditions studied. Kohl et al. [123] conducted experimental research on the flow of water and air in rectangular microchannels ($D_{h}=25-100\;\mu m$). After considering the entrance effect in liquid flow and the compressibility effect in airflow, the results were consistent with the theoretical predictions. A detailed experimental study of flow through long microchannels of hydraulic diameter ranging from 60.5 to 211 μm had been carried out by Vijayalakshmi et al. [47]. Compressibility effects and laminar-to-turbulent transition in microchannels were investigated. It has been shown that there are no special microscale effects, including an early transition to turbulence at least in the present range of hydraulic diameters after the significant effects of compressibility are appropriately accounted for. Rovenskaya et al. [124] investigated the competing effects of rarefaction and compressibility in bent microchannels. Vocale et al. [111] investigated the effect of gas compressibility on the friction factor in rectangular microchannels. The results showed that compressibility effects became increasingly stronger by reducing the hydraulic diameter and that they led to an increase in the average friction factor. In comparison to incompressible flows, for $D_{h}=500\;\mu m$, the compressibility effects increased the average friction factor up to 13% at Re = 1200; for $D_{h}=295\;\mu m$, the increase was equal to 23% at the same Reynolds number (Re = 1200). For a smaller rectangular channel ($D_{h}=100\;\mu m$), the increase was larger by reaching 20% at Re = 600. In addition to the four main scale effects mentioned above, some scholars have also investigated effects such as entrance effects [125,126,127], viscous dissipation [66,105,110,128,129], and measurement uncertainty [130,131,132].

## 3. Flow and Heat Transfer Mechanisms under the Coupling Effect of Scale Effects and Special Fluids

The application of special fluids in microchannels brings about the coupling of scale effects and changes in physical properties of the working medium, which increases the complexity of the study to a certain extent. However, most of the research in this field focuses on the unique flow and heat transfer characteristics of special fluids in microchannels rather than paying special attention to scale effects. The special flow and heat transfer characteristics of fluid with phase change (or pseudophase change) in microchannels will be described in the following sections.

### 3.1. Fluid with Phase Change in Microchannels

#### 3.1.1. Boiling Heat Transfer Mechanism

Boiling heat transfer in microchannels is widely used in industrial fields. As an efficient cooling technology, it has many advantages such as an extremely high heat transfer performance, better axial temperature uniformity, and lower pumping power consumption. Due to the presence of latent heat, flow boiling with phase change heat transfer is superior to natural convection and single-phase forced convection under high heat flux conditions [133,134]. In traditional channels, boiling heat transfer is dominated by two mechanisms: nuclear boiling and convective boiling. However, various mechanisms interact with each other in microchannels, and the development of flow patterns is difficult to predict. It is necessary to consider more influencing factors, such as bubble constraints, channel size, and shape effects. Some macroscopic heat transfer and flow phenomena may be suppressed or downgraded to secondary phenomena due to a decrease in channel size, while others may be enhanced or newly generated. This indicates that boiling in microchannels is dominated by more complex heat transfer mechanisms [135]. This article reveals the boiling heat transfer mechanism in microchannel flow from the perspective of the correlation between boiling heat transfer processes and flow patterns.

Table 5 summarizes the selected experimental studies on flow boiling heat transfer and the corresponding heat transfer mechanisms in microchannels. Researchers have conducted extensive experimental studies on flow boiling in microscale channels by changing channel diameter, working fluid type, inflow temperature, system pressure, mass flux, heat flux, etc. It is found that under the constraint of microchannel diameter, the conversion of flow modes (such as bubbly, slug, churn, wispy-annular, annular flows, etc.) is significantly affected, resulting in some flow patterns not appearing under certain operating conditions. The formation process of bubbles and the transition of flow patterns are very sensitive to the above factors. This differs from the formation mechanism of bubbles in macroscale channels.

Flow patterns and bubble dynamics are important for understanding flow boiling and two-phase flow heat transfer mechanisms in macroscale and microscale channels. Figure 16 shows the complete bubble dynamics of the microchannels [144]. Firstly, bubbles are formed at the active nucleation site, which is the starting point of the phase transition process. As bubbles grow to a certain extent and leave the nucleation site, the diameter of the bubbles determines the subsequent bubble dynamics. If the bubble diameter is relatively small compared to the channel size, the bubble moves in the flow direction after leaving the nucleation position. As the diameter of the bubble grows to match that of the channel, the growth of the bubble is restricted. Afterward, due to the continuous heating of the wall, the bubble growth process increases rapidly. Once the bubble growth is restricted by the channel cross-section, it begins to elongate along the flow direction. Elongated bubbles may eventually leave the microchannels or may cause flow reversal phenomena at high heat flux values (explained in the instability section).

Compared to conventional channels, microchannels have relatively smaller sizes, and the growth of bubbles is more easily restricted. In early studies, researchers conducted systematic research on boiling flow patterns in microchannels. Kandlikar et al. [145] suggested that the effect of surface tension was quite significant, causing the liquid to form small uniformly spaced slugs that fill the tube, sometimes forming liquid rings. Thome et al. [146] showed the strong dependency of heat transfer on the bubble frequency, the minimum liquid film thickness at dryout, and the liquid film formation thickness. Heat transfer in the thin film evaporation region was typically on the order of several times that of the liquid slug, while that for the vapor slug was nearly negligible. Chen et al. [136] observed flow patterns in small circular tubes ranging from 1.1 to 4.26 mm, including dispersed bubble, bubbly, confined bubble, slug, churn, annular, and mist flow, as shown in Figure 17. The flow characteristics in the 2.88 and 4.26 mm tubes are similar to those typically described in normal size tubes. The smaller diameter tubes, 1.10 and 2.01 mm, exhibit strong “small tube characteristics”. Magnini et al. [137] numerically simulated the effect of the primary flow parameters on the saturated flow boiling heat transfer performance of a slug flow within a horizontal microchannel. Liu et al. [138] studied the flow patterns and flow boiling characteristics in vertical rectangular microchannels. The effects of heat flux, mass flux, vapor quality, and saturation pressure on flow boiling were investigated and analyzed. For all three tested refrigerants, the heat flux and saturation pressure were found to have a significant influence on the heat transfer coefficient, while the effects of the mass flux and vapor quality were negligible. Zhang et al. [139] found that the heat transfer mechanism in small-diameter horizontal tubes is dominated by nuclear boiling. The heat transfer coefficients had little dependence on mass flux or vapor quality but had a strong correlation with heat flux and working pressure. Lv et al. [140] studied the heat transfer characteristics of the R134a/Ethane binary mixture in a horizontal microtube through numerical simulation. The simulation results showed that the boiling heat transfer coefficient increased with the heat flux and mass flow. The fluid disturbance mainly influenced the pressure drop and the turbulence intensity significantly when the fluid is transformed into a gas–liquid flow.

By combining parametric analysis with high-speed flow visualization, the relationship between heat transfer performance and flow behavior is investigated. Figure 18a,b shows schematics of axial variations of the heat transfer coefficient for flow boiling along a microchannel corresponding to two distinct heat transfer regimes. Figure 18a shows the dominant heat transfer mode of nucleate boiling, and the flow is dominated by bubbly and slug flow modes. Due to the gradual suppression of nucleate boiling, the heat transfer coefficient decreases monotonously along the channel. In contrast, Figure 18b shows the heat transfer mechanism dominated by convective boiling. Most channel lengths are dominated by annular flow, and the gradual thinning of annular liquid film leads to the increase of the heat transfer coefficient along the channel. For both heat transfer modes, when dry patches begin to form at the location of the dryout incipience, the heat transfer coefficient begins to decrease significantly [147]. Harirchian et al. [141] summarized the boiling flow patterns of a perfluorinated dielectric liquid (FC-77) at different mass fluxes and channel sizes. Various flow regimes were identified: bubbly, slug, churn, wispy-annular, and annular flows, as shown in Figure 19. Flow patterns were found to be similar for microchannels with widths of 100 μm and 250 μm and differ from those in microchannels with a width of 400 μm and larger. As the channel size increased, bubbly flow replaces slug flow, and intermittent churn/wispy-annular flow replaces intermittent churn/annular flow. Criscuolo et al. [142] studied the flow boiling heat transfer of R1234yf, R1234ze (E), and R134a in multiple microchannels. Three boiling regimes (I, II, III) were identified according to the observed effect of the heat and mass flux. Flow visualization showed that a predominantly bubbly flow was present in the channels during boiling regime I. In boiling regime II, churn flow was the predominant flow regime but appeared with more frequent vapor slugs as the heat flux was increased, eventually leading to alternating churn and wispy-annular flow. In boiling regime III, the flow regime alternated between churn and wispy-annular, with a dryout incipience leading to the progressive decrease of the two-phase local heat transfer coefficient until the achievement of the critical heat flux. The flow patterns corresponding to the three boiling regimes are shown in Figure 20. In addition, the flow direction was found to have an important effect in a study of boiling heat transfer through a microtube, which also used R134a as the working fluid [143]. It was shown that the shape of the gas slug during horizontal flow is different from the vertical direction. Heat transfer coefficient and pressure drop became increased when the refrigerant flowed in the vertical downward direction.

#### 3.1.2. Study of the Instability of Boiling Heat Transfer

The presence of bubbles during the heating application of two-phase flows leads to an increase in inhomogeneity and in some cases can even cause a reverse flow phenomenon. Flow instabilities are prevalent in flow boiling processes. Instabilities in microchannels can lead to problems such as vibrations, system control problems, and thermal fatigue and in extreme cases may even be responsible for surface burnout. Therefore, the study of two-phase flow instabilities is very important for the design and operation of microchannel heat transfer systems [134]. Ledinegg performed the first analysis of two-phase offset instability in 1938 [148]. Boure et al. [149] classified the flow boiling instability and its physical mechanism. Flow boiling instability was divided into static instability and dynamic instability. Under this classification framework, subsequent researchers have conducted systematic studies on the instability in boiling heat transfer [150,151,152,153].

Static instabilities are commonly characterized as a single-event departure from one unstable operating condition to a distinctly different operating condition [154]. This can be summarized in the following studies: critical heat flux, Ledinegg instability, boiling curve hysteresis, vapor burst, and flow pattern transition instability [155,156,157,158]. Dynamic instabilities in microchannels usually include density wave oscillations [159,160,161], parallel channel instabilities [162,163,164], pressure drop oscillations [165,166,167], acoustic oscillations [168,169], and thermal oscillations. Thermal oscillations are not an independent mode of oscillation, while the effect of acoustic oscillations is generally weak. Therefore, researchers have focused more on the other three instability modes [134]. Lee et al. [170] investigated the flow instability caused by pressure oscillation in the saturated boiling region. A flow oscillation schematic was obtained based on experimental observations, as in Figure 21. The single oscillation period was divided into six sub-periods in time, and the microchannel was divided into four distinct sections. Sub-period t1 corresponds to the initiation of the instability period; the relatively high mass velocity associated with the onset of the surge tends to suppress bubble nucleation in Section 1. Sub-period t2 is dominated by the forward advance of liquid from upstream with a slight decrease in mass velocity from peak value, while still large enough to suppress bubble nucleation upstream. Sub-period t3 is dominated by rapid bubble growth and coalescence into oblong bubbles, causing flow acceleration downstream and a slowdown upstream. Sub-period t4 corresponds to a transitional time. During sub-period t5, the vapor backflow into the inlet plenum becomes very pronounced and the bubble shape is inverted in Section 1 because of the flow reversal. Sub-period t6 marks a duration of liquid deficiency, where bubbles cast earlier into the inlet plenum mix with the plenum’s subcooled liquid, increasing the inlet liquid temperature. It has been shown that the periodic fluctuations in mass flow rate are caused by pressure oscillations between the inlet and outlet gas plenums. Barber et al. [171] suggested that the periodic pressure fluctuations across the microchannel are caused by the bubble dynamics and instances of vapor blockage during confined bubble growth in the channel. During the completely restricted growth stage of bubbles, He et al. [172] proposed an analytical model to predict pressure fluctuations by analyzing the growth of bubbles in rectangular microchannels. The interfacial velocity of the bubble was determined by solving the equations for the conservation of momentum of the liquid column and the equations of the force balance at the bubble interface, which was used to study the relationship between the reverse flow of the bubble and the pressure fluctuation. Bogojevic et al. [173] also found high-amplitude/low-frequency and low-amplitude/high-frequency instability of boiling instability in silicon microchannels. They reported that the flow pattern and type of two-phase flow instability depend on the q/G ratio and inlet subcooling conditions. The analysis of flow instabilities was conducted by examining the two-phase pressure drop, synchronous visualization results, and thermographic measurements of the channel surface temperature profiles. Wang et al. [165] identified low-frequency high-amplitude fluctuations and high-frequency low-amplitude fluctuations, which found that the main causes were periodic reverse and rewetting flow and the vapor slug cluster passage, respectively. The low-frequency high-amplitude pressure fluctuation signals and the corresponding flow pattern diagram are shown in Figure 22a,b.

### 3.2. Fluid with Pseudophase Change in Microchannels

As early as the 1930s, people began to study the heat transfer problems of fluids under supercritical pressure. The unique advantages of supercritical fluids in heat transfer have generated a lot of interest among researchers, and a lot of research has been conducted on macroscale channels [174]. In recent years, research on the application of supercritical pressure fluids in microchannels has gradually increased, but the related flow and heat transfer mechanisms still need to be further investigated. Due to the reduction of channel sizes, factors such as buoyancy, gravity, centrifugal force, and flow acceleration caused by changes in physical properties have varying degrees of influence on the flow field. Therefore, for the same supercritical fluid, the flow and heat transfer laws obtained from macrochannels are often no longer applicable to microchannels, and special forms of research are needed on the flow and heat transfer mechanism of supercritical fluid in microchannels. At present, many reviews have reported on the flow and heat transfer characteristics of supercritical pressure fluids. However, to the author’s knowledge, there are few reports on the review of the flow and heat transfer characteristics of supercritical pressure fluids in microchannels. Therefore, this article focuses on the flow and heat transfer characteristics of supercritical pressure fluids in microchannels. It also focuses on the analysis of the flow heat transfer mechanism from the perspective of the coupling effect between turbulent structure and heat transfer. Understanding and mastering the flow heat transfer mechanism of supercritical pressure fluids in microchannels under the influence of multiple factors is of great significance for the design and optimization of compact heat exchangers with supercritical work fluid as the coolant.

According to the published literature, the heat transfer process of supercritical fluids is specified in three modes: normal heat transfer, enhanced heat transfer, and deteriorated heat transfer [175,176,177,178]. Heat transfer enhancement and heat transfer deterioration may occur together. When the temperature of the thermal boundary layer reaches the pseudo-critical temperature, heat transfer enhancement will occur when the heat flow flux is low, and heat transfer deterioration will occur when the heat flow flux is high. Sometimes the transition from heat transfer enhancement to heat transfer deterioration is sudden. When heat transfer deterioration occurs, the tube wall temperature rises rapidly, which is a significant challenge to the system material and also affects system safety. Table 6 summarizes the studies related to heat transfer enhancement and heat transfer deterioration for supercritical fluids in microchannels, which have been extensively investigated by changing the boundary conditions. It can be observed that in microchannels, even if the working fluid is the same, significant differences in heat transfer phenomena will occur due to different conditions such as mass flux, heat flux, flow direction, system pressure, etc., manifested as heat transfer enhancement or deterioration. In addition, heat transfer enhancement is more likely to occur in horizontal flow [179,180], while heat transfer deterioration often occurs in vertical flow (vertical upward) [181,182,183,184]. Some of them will be explained in detail later.

#### 3.2.1. Research on Heat Transfer Enhancement Characteristics of Supercritical Pressure Fluids in Microchannels

Enhanced heat transfer is characterized by a slight increase in the wall temperature T_w_, a small temperature difference between the inner wall and the bulk, and the heat transfer coefficient is high, which mostly occurs in forced convection. A significant change in the thermophysical properties of the fluid near the pseudo-critical point leads to the maximum value of the convective heat transfer coefficient. For example, when supercritical pressure water flows in vertical and horizontal pipes, the heat transfer coefficient in the quasi-critical region increases significantly. The maximum heat transfer coefficient appears near the quasi-critical bulk enthalpy [187]. Similar results were reported by Yoshida and Mori [188]. Liao et al. [189] carried out an experimental study on the flow and heat transfer characteristics of supercritical pressure CO_2_ in small straight tubes with inner diameters of 0.70, 1.40, and 2.16 mm, respectively. The study found that the heat transfer coefficient in horizontal flow, upward flow, and downward flow differed greatly. Even though the Reynolds number of supercritical CO_2_ was up to 10^5^, the buoyance effect was still significant. Dang et al. [185] showed that the radial distribution of thermal properties in the tube cross-section is very important to interpret the experimental results. All experimental conditions showed that heat transfer enhanced when the bulk temperature of CO_2_ was near the pseudo-critical temperature. Jiang et al. [190] showed that the changes in thermophysical properties and buoyancy of supercritical CO_2_ significantly affect the convective heat transfer in vertical microtubes. Wang et al. [179] experimentally investigated the local heat transfer characteristics of supercritical carbon dioxide, which was uniformly heated in horizontal microtubes with inner diameters of 1.0 mm, 0.75 mm, and 0.5 mm. The experimental results illustrated that the heat transfer is gradually enhanced near the pseudo-critical point. Subsequently, the heat transfer coefficient increased again when the fluid temperature was sufficiently higher than the pseudo-critical value due to the enhancement of turbulence resulting from the decrease in viscosity. In addition, the experiment system exhibited the optimal heat flux with the highest thermal performance when the outlet fluid condition was close to the corresponding pseudo-critical point, at which both the specific heat and Prandtl number attained peak values.

In the experimental research on enhanced heat transfer of supercritical hydrocarbon fuels in microchannels. Gu et al. [180] conducted an experimental study on the heat transfer of supercritical cryogenic methane inside a microtube in the context of active regenerative cooling technology for liquid oxygen/methane rocket engines. Since both the Dittus-Boelter and Gnielinski correlations overestimated the measured values in the pseudo-critical region, a new semi-empirical correlation for convective heat transfer of supercritical cryogenic methane based on PDF was developed, as shown in Figure 23a,b. This was similarly in the experimental research on active regenerative cooling. Votta et al. [191] represented the cooling channel of a typical rocket engine by using a rectangular cross-section of a microchannel, with cryogenic methane as the coolant and fuel at supercritical pressure. The maximum heat flux at the bottom of the channel can reach 20 MW/m^2^. In particular, the effect of channel surface roughness caused by the manufacturing process was evaluated, and a Nusser number correlation for heat transfer in rectangular channels was established. Li Hui et al. [186] systematically investigated the effects of different boundary conditions such as heat flow flux, mass flow rate, and system pressure on the heat transfer characteristics of supercritical methane in a microchannel through a combination of experiments and numerical simulations.

#### 3.2.2. Research on Heat Transfer Deterioration Characteristics of Supercritical Pressure Fluids in Microchannels

Heat transfer deterioration is characterized by a sharp increase in wall temperature T_w_, a large temperature difference between the inner wall and the bulk, and a sudden drop in the heat transfer coefficient, which mostly occurs in mixed convection. The variation of wall temperature along the flow direction when heat transfer deterioration occurs is shown in Figure 24a,b [192]. Hall and Jackson [193] provided a theoretical explanation for the mechanism of heat transfer deterioration, showing that the dominant factor is a change in the shear stress distribution over the channel, with corresponding changes in turbulence generation. In particular, Tanaka et al. [194] investigated the effect of buoyancy and flow acceleration on the turbulent forced convective heat transfer of supercritical fluid in a vertical pipe. This study demonstrated that the effects of the two forces are very similar, leading to a rapid decrease in shear stress near the wall. They also derived criteria for the significant effects of buoyancy and acceleration, and proposed verse transition criterion from turbulence to laminar flow. Jackson and Hall [195,196] later provided general criteria for the onset of buoyancy effects applicable to supercritical pressure and subcritical pressure fluids.

With the comparison of experimental results for upward flow with those for downward flow under otherwise identical conditions, the explanation of special heat transfer phenomena induced by buoyancy and flow acceleration of fluids at supercritical pressure became more accepted by scholars [193,197]. The buoyancy effect may lead to a deterioration or enhancement of heat transfer in upward flow, but heat transfer in downward flow often increases, while the acceleration effect may lead to a deterioration of heat transfer in both upward and downward flow. In general, the buoyancy effect originates from the radial gradient of density, while the flow acceleration effect originates from the axial gradient of density. Lei et al. [181] found that at low q/G (the ratio of heat fluxes to mass fluxes), there is an apparent heat transfer enhancement and insignificant difference in the two arrangements. However, when the q/G increases to a higher value (i.e., q/G > 0.5), heat transfer deterioration occurs and a noticeable heat transfer discrepancy is detected, where the inner wall temperature of vertical flow far exceeds that of horizontal flow, as shown in Figure 25a,b. When using the q/G as a criterion to determine the appearance of heat transfer deterioration in supercritical pressure fluids, Peng et al. [182] gave a different conclusion from their experimental study. They found that at low mass flux, no significant heat transfer deterioration behavior was observed even if the q/G was high. However, significant heat transfer deterioration occurred under the same value of q/G with high mass flux. The reason for this is that at low mass flow rates, strong buoyancy and significant acceleration effects enhance heat transfer, but at high mass flow rates, heat transfer deterioration is mainly influenced by buoyancy rather than acceleration effects. Therefore, only using the q/G as a criterion for the occurrence of heat transfer deterioration often fails under certain operating conditions. Liu et al. [183] conducted an experimental study on the convective heat transfer of n-decane at supercritical pressure in vertical tubes with inner diameters of 0.95 and 2.00 mm, respectively. The effects of heat flow flux, property variation, buoyancy, and flow acceleration on heat transfer were investigated. Similarly, for n-decane under supercritical pressure, Zhu et al. [184] showed that buoyancy can affect the intensity of turbulent kinetic energy, which in turn affects the heat transfer near the wall. When the local buoyancy was less than the threshold, the effect of buoyancy on heat transfer could be ignored. After exceeding the threshold, buoyancy promoted heat transfer in the downward flow structures but weakened heat transfer in the upward flow structures. Similar conclusions were reached by Zhao et al. [198] in their experimental study of convective heat transfer for RP-3 at supercritical pressure. Bao et al. [199] studied the flow resistance characteristics of n-decane in a vertical microtube under supercritical pressure. The effects of heat flux and system pressure on pressure drop are shown in Figure 25a,b. Based on the experimental results, an empirical correlation for the friction coefficient was summarized for different temperatures, heat fluxes, and mass flow velocity ratios.

#### 3.2.3. Turbulent Structure and Heat Transfer Coupling Mechanism

The existence of near-wall turbulent structures has been studied in detail for decades, and their structural characteristics have continued to deepen the understanding of the coexistence of turbulent randomness and orderliness [200]. The typical organizations of the outer layer of the turbulent boundary layer are large-scale structures, as shown in Figure 26 [201]. Large-scale structures play an important role in turbulent transport in the outer boundary layer, thickening the boundary layer by entraining the external non-swirling flow and inducing near-wall instability. The inner near-wall turbulence structure and the outer layer structure are not independent, but they interact with each other. The classical theory of near-wall turbulent structure is also applicable to the flow in microchannels. Therefore, understanding the formation, evolution, and interaction of turbulent structures is particularly important for revealing the turbulent flow and heat transfer mechanism of supercritical fluids in microchannels. Yoo [202] also suggested that there is no consensus on the general trends of heat transfer in supercritical fluids, especially in terms of turbulence, and highlighted the importance of studying the turbulent structure of supercritical fluids in depth.

Numerical calculations conducted using direct numerical simulation (DNS) or large eddy simulation (LES) are considered effective methods for obtaining accurate data on heat transfer and detailed information about flow structure. You et al. [203] conducted a DNS study on turbulent mixed convection in a vertical circular tube. The influence of buoyancy on the turbulent transport of momentum and heat was elucidated by the ‘external’ and ‘structural’ effects. The first DNS study of the flow and heat transfer characteristics of a fluid at supercritical pressure in a vertical micropipe was carried out by Bae et al. [204]. It was demonstrated that the DNS results are in good agreement with the average velocity, turbulence intensity, and Nusselt number obtained by McEligot’s [205] experiments. The detailed information about the flow field and thermal field can provide a deeper understanding of the physical mechanisms of turbulence reduction and recovery, which helps to establish a general understanding. The changes in turbulent kinetic energy generation terms are shown in Figure 27a,b. Subsequently, many scholars have confirmed the feasibility of using DNS or LES calculations in supercritical pressure fluid [206,207,208,209,210,211].

In fluid heat transfer at supercritical pressure, wall heat transfer significantly changes the velocity fluctuations and turbulent structures near the wall, which in turn affects the local heat transfer performance. It has been found that transcritical temperature conditions enhance heat transfer fluctuations and alter the turbulence generation rate in wall-bounded flow [202]. This deviation from the behavior of ideal gas should not be confused with the real fluid effect, which refers to the molecular dissociation gas occurring in hypersonic flow. For the supersonic boundary layer with an adiabatic wall, it is found that the dynamics of the near-wall turbulent structure is similar to the case of constant physical properties and conforms to the classical scale based on the wall. For cooling walls, the near-wall streaks extend along the flow direction, while for heating walls, the near-wall streaks shorten along the flow direction [212]. According to Sciacovelli et al.’s [213] DNS of dense gas, supersonic turbulent channel flow, transport properties are dependent on the density and temperature of the fluid, and due to the real fluid effect, the sound velocity varies non-monotonically. The dense gas effect caused the maximum root mean square of the density fluctuation to be located in the viscous sublayer, which was different from the ideal gas case where it was located in the buffer layer. Therefore, density fluctuation does not significantly alter the turbulent structure in the channel, and Morkovin’s hypothesis remains valid.

However, the turbulent behavior of supercritical pressure fluids that also causes density fluctuations is different. Because the fluid temperature near the wall first crosses the critical value under strong heating conditions, there is a sharp change in physical properties within the narrow temperature zone near the quasi-critical temperature, and the peak density fluctuations are located in the buffer layer rather than the viscous sublayer. Kawai et al. [214] found that Morkovin’s hypothesis (Morkovin 1962) is not applicable to pseudophase transition conditions. Due to significant density fluctuations, non-classical effects were generated in mass flux, turbulent diffusion, and pressure expansion distributions. By studying the pseudophase transition effect in turbulent channel flow under transcritical conditions [215], it was found that the peak of density fluctuations during the pseudophase transition process does not exist in the viscous sublayer but in the buffer layer. The pseudophase transition effect had a direct impact on the turbulent structure near the wall. The apparent ejection event from the liquid-like region (near the cold bottom wall) lifted the high-density fluid into the low-density core of the channel. Due to the greater inertia of the ejected high-density fluid (the study did not consider gravity effects), it reached the core of the channel, where the fluid underwent a pseudophase transition, effectively achieving mass transfer. As shown in Figure 28, large-scale flow-aligned structures are observed near the bottom wall (see circles). These structures are spatially correlated with the occurrence of turbulent ejection events.

The significant variation of velocity fluctuation is also attributed to the large density fluctuation caused by the special pseudophase transition phenomenon under the transcritical condition. More specifically, it is caused by the sudden transition from a liquid high-density fluid to a gaseous low-density fluid through the quasi-critical temperature Tpc in a narrow temperature range, and the effect of near-wall velocity fluctuations is directly reflected in the turbulent burst behavior. Turbulent burst is usually studied by the quadrant analysis method, where Q2 and Q4 motions are associated with ejection and sweep events, while Q1 and Q3 motions are referred to as outward and inward interactions. In the flat plate turbulent boundary layer where zero pressure gradient of fluid is fully developed under supercritical pressure, turbulent ejection (Q2) and turbulent sweep (Q4) are changed [214]. In the transcritical boundary layer, the velocity fluctuation corresponding to Q2 events increases significantly, while the velocity fluctuation corresponding to Q4 events decreases. This is significantly different from the wall turbulence behavior in the cold state, where the velocity fluctuation intensity of Q2 and Q4 events near the wall is basically the same.

The turbulent structures corresponding to the above Q2 and Q4 turbulent behavior are shown in Figure 29 [214], where narrow low-speed and high-speed streaks with typical flow elongation are observed in the near-wall region. The low-speed streaks are accompanied by the upward (v″ > 0) vertical transport of a heated gas-like low-density fluid (T > T_pc_) near the wall, which is related to the Q2 ejection event and characterized by u″ < 0, v″ > 0, and $\rho^{\prime \prime }<0$. The high-speed streaks are accompanied by the downward (v″ < 0) vertical transport of the upper unheated liquid high-density fluid (T < T_pc_), which is related to the Q4 sweep event and characterized by u″ > 0, v″ < 0, and $\rho^{\prime \prime }>0$. In the heated transcritical turbulent boundary layer, the streaks structure with Q2 ejection and Q4 sweep dominates the generation of Reynolds shear stress (and the generation of turbulent kinetic energy). The mechanism of significant changes in velocity fluctuations is believed to be caused by the rapid expansion and compression process of fluids caused by density fluctuations within a narrow temperature range of transcritical conditions.

After affirming the significant impact of density fluctuations on flow structures, some scholars further emphasize that the impact of dynamic viscosity fluctuations is equally important. In early research on turbulence with temperature-dependent dynamic viscosity, the effect of wall heating on linear stability in laminar shear flow with temperature-dependent viscosity was analyzed [216,217,218,219]. The above studies have addressed the effect of heating on transition, but much is still unknown about the effects of fully developed turbulence, such as turbulent surface friction and flow structures. Lee et al. [220,221] studied the effect of viscous stratification caused by wall heating on the friction reduction of the turbulent boundary layer. Within the simulated Reynolds number range, the surface friction coefficient could be reduced to 26% of its reference value. It was shown that the reduced Reynolds shear stress and the weakened mean momentum near the wall are the main reasons for the reduced friction coefficient at the wall. The role of turbulent structures was mainly discussed, and the instantaneous three-dimensional vortical structures are visualized in Figure 30. However, in this study, only the dynamic viscosity changed with temperature, while other physical properties remained constant. Therefore, the author emphasized the necessity of introducing buoyancy, thermal diffusion, and other effects when studying other physical properties that also change.

The flow and heat transfer process under supercritical pressure belongs to the research scope emphasized by Lee. Not only does the dynamic viscosity change with temperature but also other physical properties change with temperature and pressure and become more complex with the appearance of quasi-critical states. For example, in the study on the turbulent attenuation of CO_2_ in a heated and cooled annular flow under supercritical pressure, the reduction (or increase) of turbulent motion was attributed to the transient density and dynamic viscosity fluctuations of the heated (or cooled) fluid under supercritical pressure. The turbulent shear stress and turbulence intensity decreased significantly near the hot inner wall but increased near the cold outer wall, which resulted in a decrease in the production of turbulent kinetic energy near the inner wall and an increase near the outer wall. By analyzing the transport equation of coherent streak flank strength, it was found that fluctuations in thermophysical properties significantly affect streak evolution. Thermal expansion and buoyancy tend to reduce streaks coherence near the hot wall. However, near the cold wall, the results were the opposite: the coherent streaks flank strength and the flowing vorticity were enhanced by the density and dynamic viscosity changes, as shown in Figure 31. In a further study [222], the effects of specific heat, thermal diffusion, density, and molecular Prandtl number changes on turbulent heat transfer were analyzed. The study showed that specific heat capacity fluctuations are the main reason for temperature fluctuations. Compared to the enthalpy fluctuations, temperature fluctuations were enhanced in regions with low specific heat capacity and diminished in regions with a large specific heat capacity. By analyzing the Nusselt numbers under different conditions, it was found that heat transfer deterioration or enhancement can occur without streamwise acceleration or mixed convection conditions. The typical flow structures near the hot wall of the annular tube are shown in Figure 32 and Figure 33.

The research on turbulent thermal boundary layer showed that under isothermal and isoflux wall boundary conditions, the temperature distribution and flow structure distribution near the wall of constant physical fluid still maintains a strong similarity [224]. However, the effect of thermal boundary conditions on supercritical pressure fluids was very different and was mainly reflected in the influence of parameter fluctuations near the wall. When using isothermal and isoflux wall boundary conditions to control wall temperature fluctuations, and then studying the influence of wall temperature fluctuations on the heat transfer mechanism of supercritical fluid flow [209], it was found that for fluids with constant physical properties (Pr > 1), the influence of wall temperature fluctuations on the average enthalpy and Nusselt number distribution was very limited. Compared with fluids with constant physical properties, the heat transfer of supercritical fluids with Pr > 1 strongly depended on the hot wall boundary conditions. If thermal fluctuations were allowed on the wall, a significant increase in Nusselt number and mainstream enthalpy was observed. In addition, wall temperature fluctuations could cause dramatic fluctuations in density, viscosity, and thermal conductivity and increased turbulent shear stress and turbulent heat transfer. When the wall temperature did not fluctuate, the turbulent shear stress and turbulent heat flow would be attenuated. The comparison of instantaneous enthalpy fluctuations under the two boundary conditions is shown in Figure 34. The simulation of compressible channel flow under supercritical pressure has been studied by Sengupta et al. [225]. Their results showed that the cold wall region is characterized by much higher occurrences of high-temperature and high-density fluctuations than the region near the hot wall. In addition, as the semi-local Reynolds number increased, the characteristics of the liquid-like flow region decreased in the streamwise anisotropy and increased in the spanwise anisotropy, while the opposite was seen with the gas-like flow.

On the other hand, studies on the coupling of turbulent structures with heat transfer focus on the effects of buoyancy and/or flow acceleration, since these factors lead to a decrease in turbulence intensity and consequently decrease turbulence mixing. The effects of buoyancy and acceleration can be considered as indirect effects of variable physical properties on heat transfer by changing the flow field and thus affecting convective heat transfer. Kurganov et al. [226] showed that the development of tendencies toward the deterioration of heat transfer is associated with the deep reconstruction of the fields of velocity and shear stresses under the combined influence of buoyance forces and the negative pressure gradient that accelerates the heated fluid flow. Nemati et al. [227] found that under the condition of the low buoyancy effect, thermal expansion caused by isothermal wall boundary conditions attenuates turbulent kinetic energy. However, enhanced turbulence was observed under high buoyancy conditions. Cao et al. [228] used direct numerical simulations to investigate the effect of coupled buoyancy and thermal acceleration on the turbulent flow and heat transfer of supercritical CO_2_ in a vertical pipe at a low Reynolds number. The results showed that the turbulent flow and heat transfer under the effect of buoyancy and thermal acceleration present four developmental periods, as shown in Figure 35, in which buoyancy and temperature acceleration are alternately dominant. The criterion for distinguishing heat transfer deterioration under complex coupling of buoyancy and thermal acceleration was proposed. Furthermore, the analysis of the orthogonal decomposition and generation mechanism of turbulent structures indicated that the significant flow acceleration can destroy the three-dimensional flow structure and stretch the vortices resulting in dissipation.

Considering the huge cost of DNS in terms of computational resources and time consumption, the large eddy simulation numerical method is considered to be one of the best choices for the compromise between computational accuracy and computational volume. The application of large eddy simulation has been involved in the early study of flows with variable properties. Wang and Pletcher [229] carried out a large eddy simulation of compressible turbulence in 1996, and the fluid properties changed greatly, but the buoyancy effect was not considered in this study. In a study of mixed convection considering buoyancy, Lee et al. [230] conducted large eddy simulations on horizontal channels with significant heat transfer. The results showed that as the Grashof number increased and large-scale turbulent motion appeared near the wall, leading to significant changes in the turbulence intensity of the horizontal channel flow. Although the above research considers the change of thermophysical properties, it does not belong to the category of supercritical fluids. As a reliable method to explore the mechanism of flow and heat transfer of supercritical fluids in microscale channels, LES technology has been rapidly applied to supercritical fluids in recent years.

Wang [208] (2021) emphasized that his research was the first effort to study the heat transfer of supercritical water with LES methodology, using the opensource CFD code OpenFOAM rather than commercial software. Xie et al. [231] (2022) reproduced a series of DNS results through the large eddy simulation method, emphasizing that the LES results had a good consistency with DNS data, and the heat transfer deterioration and recovery process could be well captured. In the study of upward flow in a micropipe with a diameter of 2 mm, they found that the mean velocity profile is distorted into an M-shaped distribution, corresponding to the regeneration of turbulent kinetic energy and Reynolds stress. The transient turbulent streaks and vortex structures are shown in Figure 36 and Figure 37, which help to understand the development of turbulent structures along the pipe. Similarly, in a vertical circular tube, a large eddy simulation study of supercritical pressure water found that compared to forced convection, the heat transfer during downward flow was enhanced, while the heat transfer during upward flow was significantly damaged [232]. The difference in heat transfer characteristics was attributed to the influence of buoyancy, which could be demonstrated by analyzing the turbulent statistical distribution of the flow pattern. Tao et al. [233] (2018) demonstrated a large eddy simulation method for hydrocarbon fuel flowing through a microtube of uniform heating under supercritical pressure. Zhang et al. [234] (2022) conducted a large eddy simulation study on the heat transfer mechanism of supercritical pressure methane in a horizontal micropipe (D = 0.8 mm), taking into account the effects of physical property fluctuations and buoyancy on the near-wall turbulent structure. An in-depth analysis of the heat transfer mechanism from the perspective of near-wall turbulent burst events was also presented. In order to improve the thermal performance of regenerative cooling channels for advanced aircraft, Sun et al. [235] (2021) used large eddy simulation methods to study the flow behavior, vortex structure, and transport characteristics of supercritical n-decane in microchannels. Flow patterns were determined by several common buoyancy criteria, and oscillatory effects were observed in strongly mixed flows. The results showed that buoyancy can cause thermal oscillations that redistribute the flow structure.

#### 3.2.4. Heat Transfer Correlations

For decades, extensive research has been conducted on the convective heat transfer of various supercritical fluids, resulting in a large amount of experimental data. Various heat transfer correlations have been established based on dimensionless parameters and property models, and the majority of these correlations take the following form [236]:(4)Nu=k1Rek2Prk3ρwρbk4cp¯cpbk5μwμbk6λwλbk7

Here, k_1_–k_7_ are constants, and the significant variation of fluid properties is reflected in the comparison of properties obtained based on the reference temperature of wall temperature and the reference temperature of bulk temperature. The detailed descriptions of some correlations in microchannels are summarized in Table 7. Unlike the Dittus-Boelter’s heat transfer correlation (Nub=0.023Reb0.8Prb0.4), which has good applicability for fluids of constant physical properties in channels, there appears to be no universal correlation for supercritical fluids in channels. As can be seen from Table 7, these correlations can only be applied to certain specific fluids and conditions, such as in Dang [185], Wang [179], and Gu [180], which are only applicable to horizontal flow, or in Peng [182], Liu [183], and Fu [237], which are only applicable to vertical flow. If the range of correlations is ignored, their accuracy in predicting heat transfer will not be guaranteed or even give incorrect results, so particular care should be taken when using specific correlations.

In general, the form given in Equation (4) can still be used as the base form for the heat transfer correlations of supercritical pressure fluids in microchannels. According to different operating conditions, researchers can add buoyancy, thermal acceleration, flow patterns, dimensionless heat flux, and other factors into the heat transfer correlations to improve the prediction accuracy of the correlation in the corresponding application fields. Furthermore, in order to establish heat transfer correlations for supercritical pressure fluids with wider applicability, the development of new correlations should pay more attention to physical processes such as heat transfer modes and flow structures, which remains one of the key tasks in this research field.

## 4. Conclusions

This article systematically reviews the microscale flow and heat transfer research, focusing on the flow and heat transfer mechanisms in microchannels. The microscale flow and heat transfer mechanisms under the influence of multiple factors, including scale effects (such as rarefaction, surface roughness, axial heat conduction, and compressibility) and special fluids, are investigated, which can meet the specific needs for the design of various microscale heat exchangers. The results can be summarized as follows:(1)The current research on flow and heat transfer characteristics in microchannels can be divided into two categories: one is the microscale single-phase flow and heat transfer, which excludes the effect of fluid physical properties changes and focuses on the mechanism of the scale effects due to the reduction of the channel size. The other is the application of special fluids (fluid with phase change (pseudophase change)) in microchannels, which brings about the coupling of the scale effects with the change of the physical properties, and the overall heat transfer performance is further enhanced. However, most of the studies in this field have focused on the flow and heat transfer characteristics of special fluids in microchannels, rather than on the scale effects in particular.(2)For the flow and heat transfer mechanisms of single-phase fluid in microchannels, it is first necessary to determine whether the flow satisfies the continuity assumption by the *Kn* number. If *Kn* < 0.001, the flow is regarded as continuous. Although the basic equations and physical laws at conventional scales are applicable, scale effects (rarefaction effects, surface roughness, wall axial heat conduction, compressibility, etc.) must be taken into account, which is an important reason for the deviation of macroscopic classical flow and heat transfer theories in microchannel single-phase flow. If *Kn* > 0.001, the assumption of continuity no longer holds and rarefaction effects become significant. It is necessary to modify the boundary conditions (slip velocity, temperature jump) or apply the molecular dynamics theory to solve the difference between classical continuum flow and molecular transport flow.(3)The flow boiling heat transfer mechanism in microchannels is very different from that in conventional channels. In the boiling heat transfer process of microchannels, the development of bubbles goes through a series of dynamic processes that include bubble nucleation, growth, departure, and movement along the flow direction. However, the development patterns of conventional scale flow patterns are often unable to predict flow pattern transitions in microchannels. It is necessary to consider more influencing factors, such as bubble constraints, channel size, and shape effects. The above factors play an important role in flow heat transfer characteristics, as well as various flow instabilities.(4)For the flow and heat transfer mechanism of the fluid with pseudophase change in microscale channels, the significant change of physical properties in the transcritical process leads to dramatic velocity fluctuations, which directly affects the turbulent burst behavior of the flow field (ejection event Q2 and sweep event Q4). Buoyancy and thermal acceleration are considered to be indirect effects on the flow field, which may lead to the attenuation of turbulence intensity and reduce the ability of turbulent heat transfer. In addition, the mechanisms of different heat transfer modes for supercritical pressure fluids in microchannels have not been unified. In particular, how to maximize the performance of heat transfer enhancement while effectively avoiding heat transfer deterioration remains to be further studied.

## Figures and Tables

**Figure 1 micromachines-14-01451-f001:**
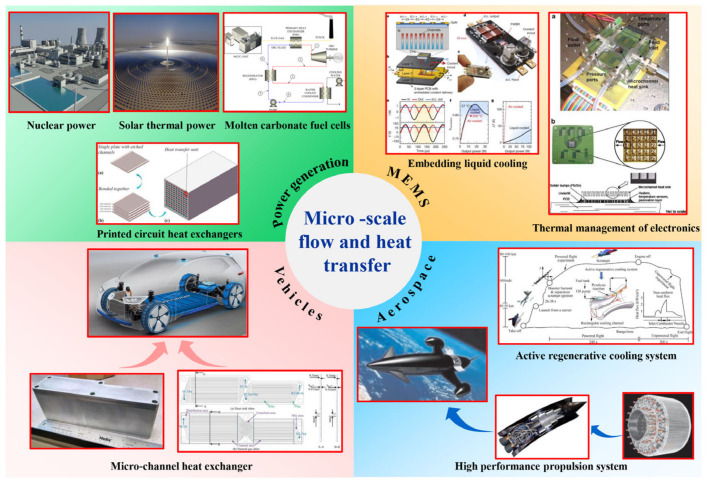
Applications of microscale flow and heat transfer in industrial fields.

**Figure 2 micromachines-14-01451-f002:**
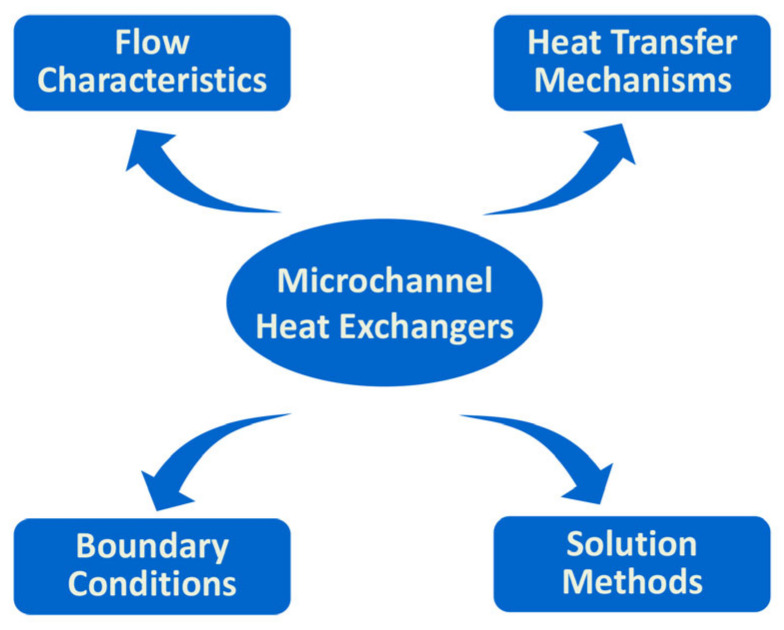
Classification diagram of microchannel heat exchangers.

**Figure 3 micromachines-14-01451-f003:**
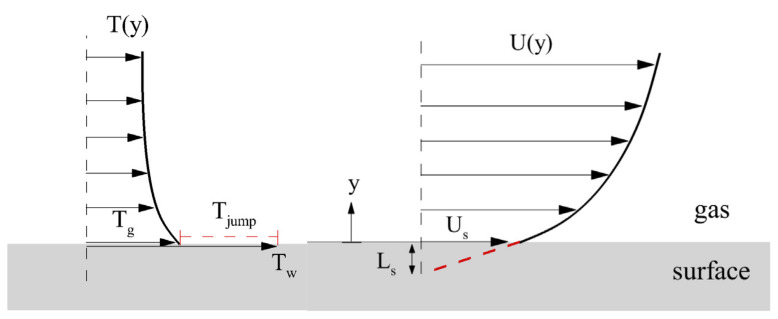
Velocity slip ($U_{s}$), slip length ($L_{s}=U_{s}/\left(\mathrm{dU}/\mathrm{dy}\right)$), and temperature jump ($T_{\mathrm{jump}}$) in the presence of non-equilibrium gas–surface interactions in rarefied gas flows [51]. Reprinted with permission from Ref. [51], *Physics Reports*, Copyright © 2022 Elsevier B.V.

**Figure 4 micromachines-14-01451-f004:**
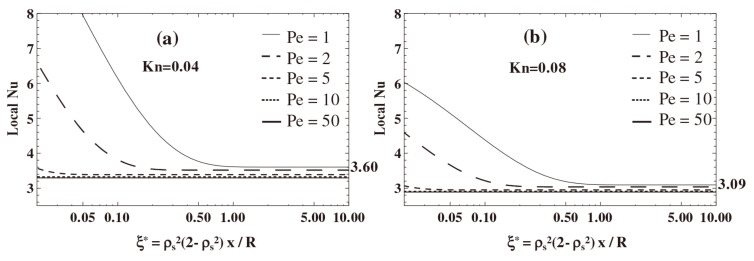
Variation of local Nusselt number along $\xi^{*}$ for different Peclet number values at (**a**) *Kn* = 0.04 and (**b**) *Kn* = 0.08 [52]. Reprinted with permission from Ref. [52], *International Communications in Heat and Mass Transfer*, Copyright © 2015 Published by Elsevier Ltd.

**Figure 5 micromachines-14-01451-f005:**
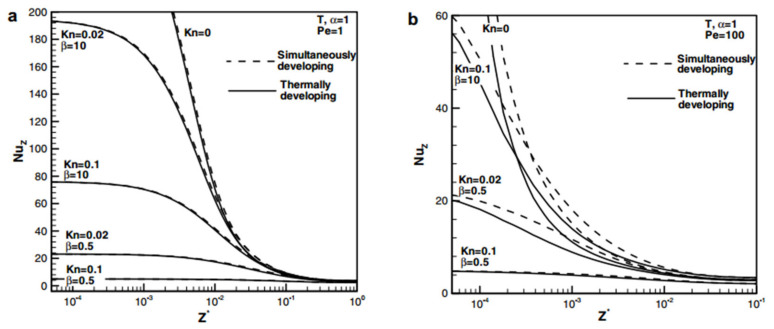
Effect of *Kn* and β on the developing local Nusselt number along the axis of a square microchannel for (**a**) *Pe* = 1; (**b**) *Pe* = 100 [63]. Reprinted with permission from Ref. [63], *International Journal of Heat and Mass Transfer*, Copyright © 2008 Elsevier Ltd.

**Figure 6 micromachines-14-01451-f006:**
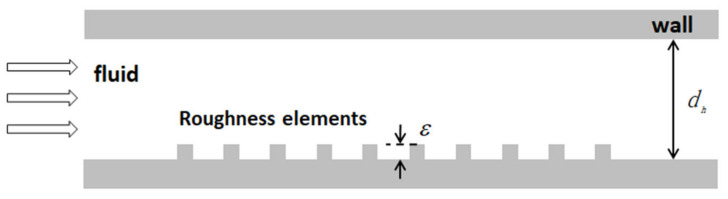
Schematic diagram of surface roughness in microchannels.

**Figure 7 micromachines-14-01451-f007:**
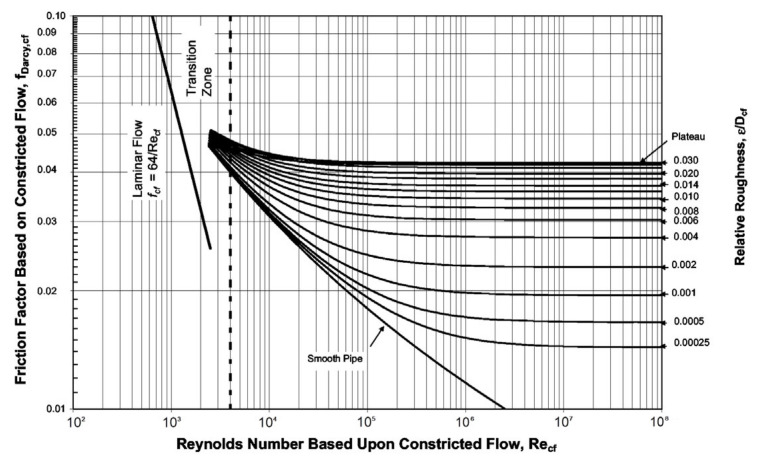
Darcy friction factor representation based on the constricted flow diameter [69]. Reprinted with permission from Ref. [69], *Physics of Fluids*, Copyright © 2005 Published by AIP Publishing.

**Figure 8 micromachines-14-01451-f008:**
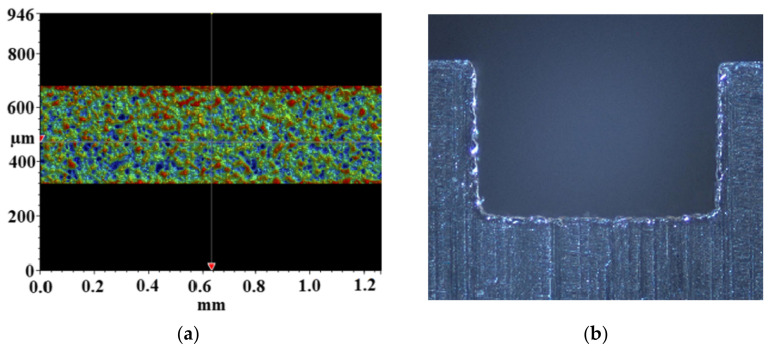
The distribution of surface roughness in microchannels: (**a**) surface maps taken from the optical profilometer; (**b**) results taken from the optical microscope [42]. Reprinted with permission from Ref. [42], *Applied Thermal Engineering*, Copyright © 2022 Elsevier Ltd.

**Figure 9 micromachines-14-01451-f009:**
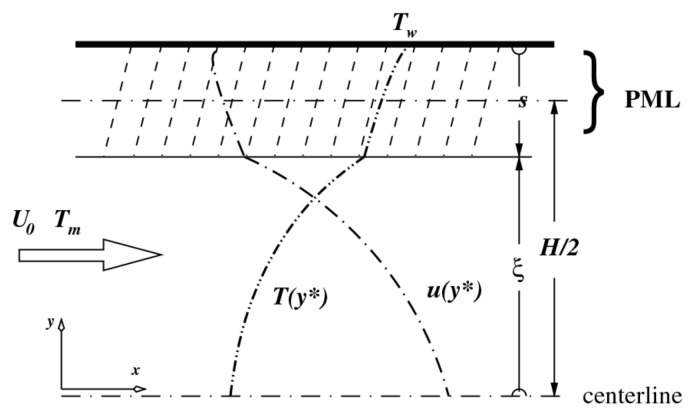
Schematics of the porous medium layer (PML) [79]. Reprinted with permission from Ref. [79], *International Journal of Heat and Mass Transfer*, Copyright © 2005 Elsevier Ltd.

**Figure 10 micromachines-14-01451-f010:**
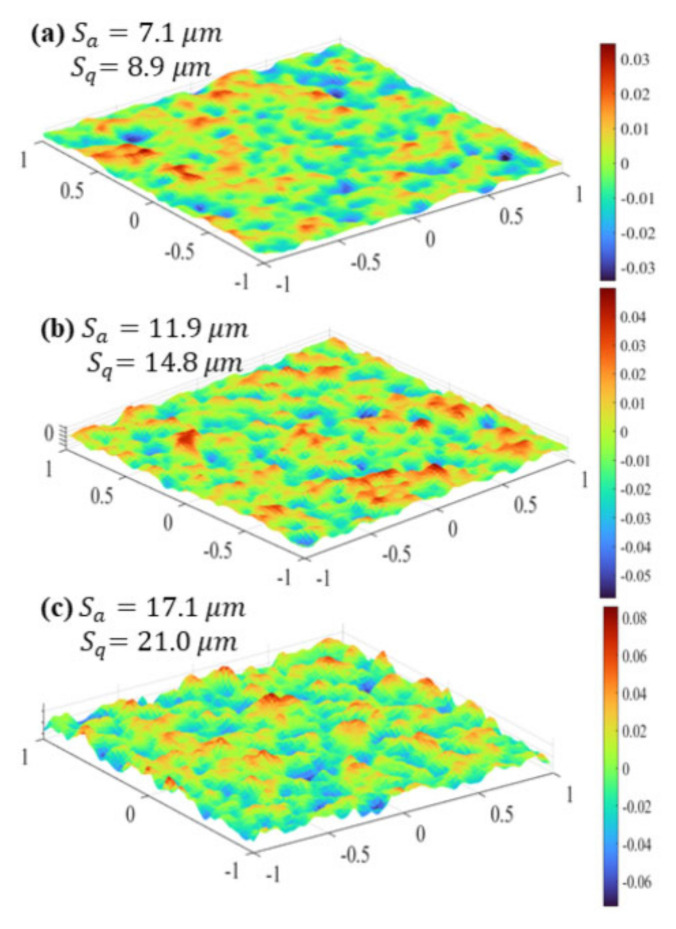
Rough mesh surfaces generated with ACL. (**a**) RMC1, (**b**) RMC2, (**c**) and RMC3 [82]. Article published with open access.

**Figure 11 micromachines-14-01451-f011:**
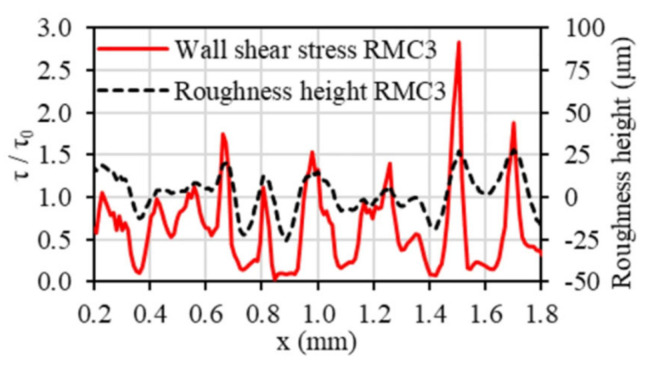
Variation of local viscous shear stress ratio with roughness profile [82]. Article published with open access.

**Figure 12 micromachines-14-01451-f012:**
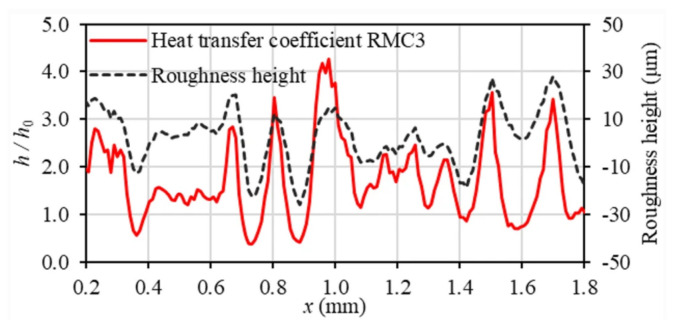
Variation of local heat transfer coefficient ratio with roughness profile [82]. Article published with open access.

**Figure 13 micromachines-14-01451-f013:**
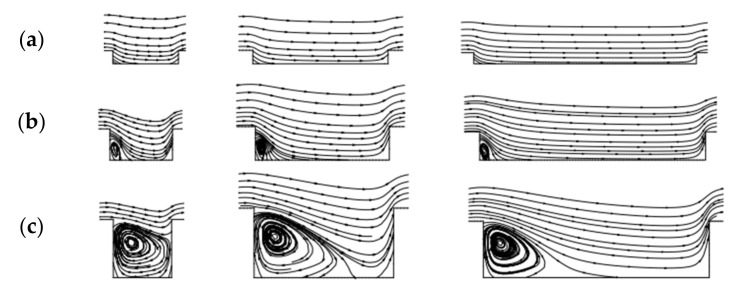
Flow patterns near roughness units under different relative roughness (k/D): (**a**) k/D = 0.01; (**b**) k/D = 0.025; (**c**) k/D = 0.05; and (**d**) k/D = 0.1 [90]. Reprinted with permission from Ref. [90]. *The Journal of Fluid Mechanics*, Copyright © 2019 Cambridge University Press.

**Figure 14 micromachines-14-01451-f014:**
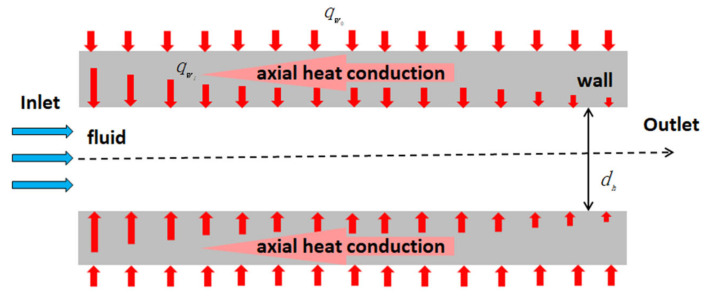
Schematic diagram of axial thermal conductivity.

**Figure 15 micromachines-14-01451-f015:**
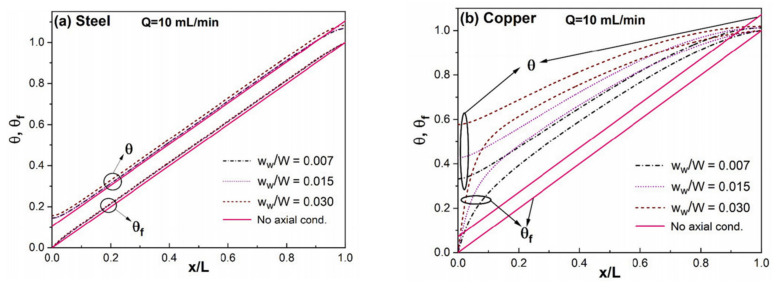
Axial conduction induced drift in the wall (θ) and fluid (θf) temperature profiles as a function of (x/L): (**a**) steel and (**b**) copper channels for different wall thicknesses [102]. Reprinted with permission from Ref. [102], *International Communications in Heat and Mass Transfer*, Copyright © 2021 Elsevier Ltd.

**Figure 16 micromachines-14-01451-f016:**
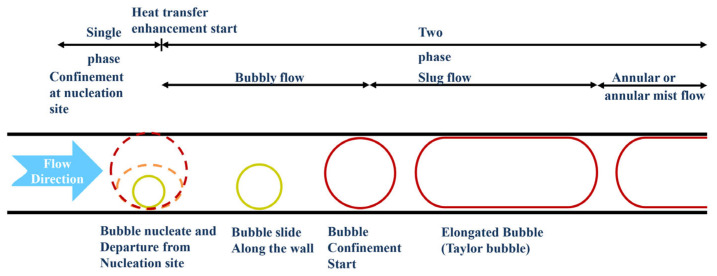
The dynamic process of bubble development in microchannels [134]. Reprinted with permission from Ref. [134], *Applied Thermal Engineering*, Copyright © 2022 Elsevier Ltd.

**Figure 17 micromachines-14-01451-f017:**
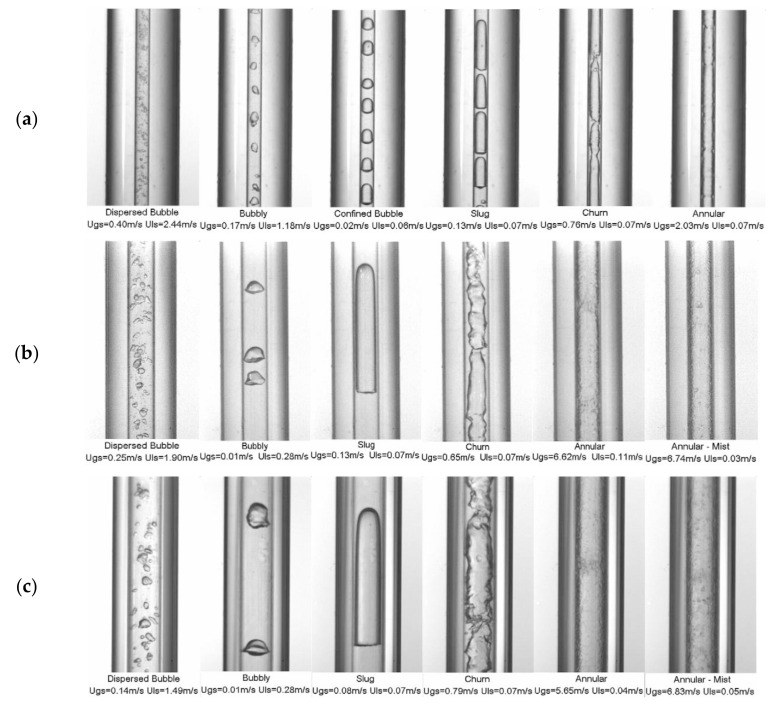
Flow patterns observed in the different internal diameters tube at 10 bar: (**a**) 1.1 mm; (**b**) 2.01 mm; (**c**) 2.88 mm; and (**d**) 4.26 mm [136]. Reprinted with permission from Ref. [136], *International Journal of Heat and Mass Transfer*, Copyright © 2006 Elsevier Ltd.

**Figure 18 micromachines-14-01451-f018:**
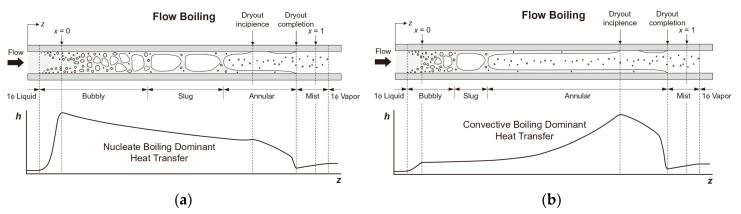
Schematics of flow regimes and variation of heat transfer coefficient in mini/microchannels with uniform circumferential heat flux: (**a**) nucleate boiling dominant heat transfer; (**b**) convective boiling dominant heat transfer [147]. Reprinted with permission from Ref. [147], *International Journal of Heat and Mass Transfer*, Copyright © 2014 Elsevier Ltd.

**Figure 19 micromachines-14-01451-f019:**
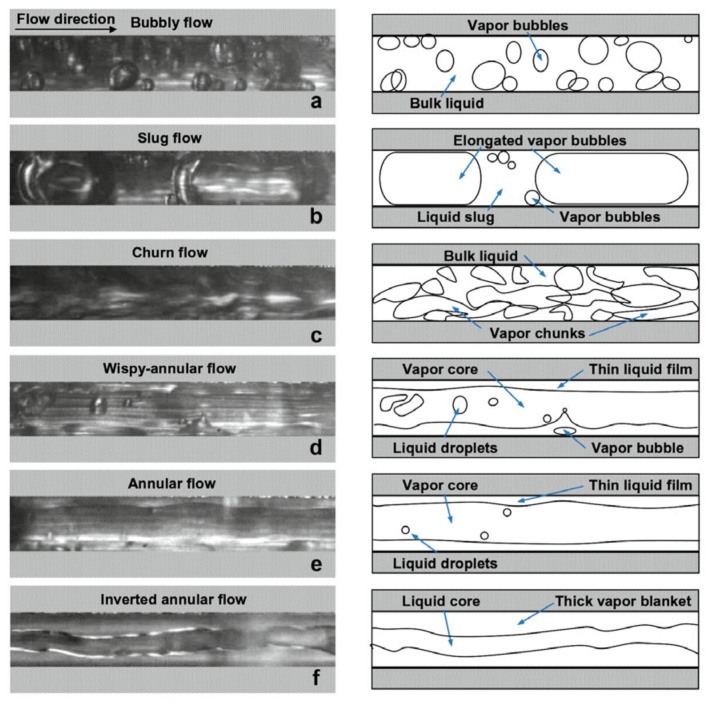
Description of different boiling flow regimes: (**a**) Bubbly flow; (**b**) Slug flow (**c**) Churn flow (**d**) Wispy-annular flow (**e**) Annular flow (**f**) Inverted annular flow [141]. Reprinted with permission from Ref. [141], *International Journal of Multiphase Flow*, Copyright © 2009 Elsevier Ltd.

**Figure 20 micromachines-14-01451-f020:**
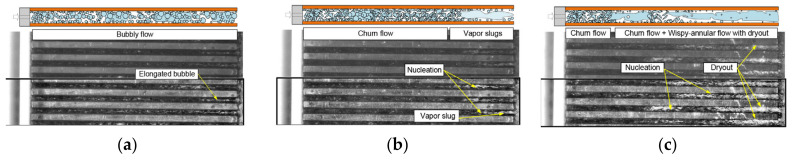
Flow patterns corresponding to three boiling regimes [142]. Article published with open access; (**a**) x_out_ = 0.12, q_w_ = 17 W/cm^2^, boiling regime I; (**b**) x_out_ = 0.53, q_w_ = 67 W/cm^2^, boiling regime II; (**c**) x_out_ = 0.78, q_w_ = 94 W/cm^2^, boiling regime III.

**Figure 21 micromachines-14-01451-f021:**
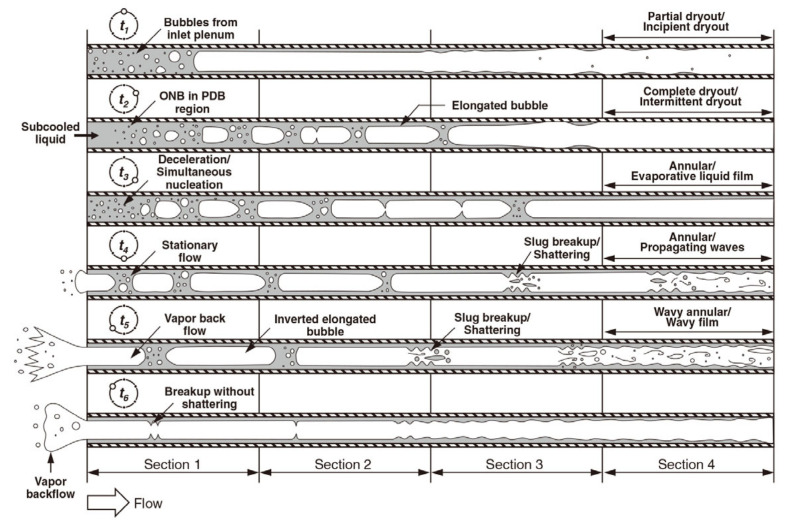
Schematic renderings of transient flow patterns observed at six different times within a single periodic cycle: *t*_1_: beginning of the cycle, *t*_2_: forward liquid advance, *t*_3_: rapid bubble growth, t4: commencement of transition pattern, *t*_5_: commencement of wavy annular pattern, and *t*_6_: end of liquid deficient period. The indicated sections are as follows: 1: subcooled boiling dominant, 2: combined saturated nucleate and convective boiling, 3: saturated convective boiling by annular film evaporation, and 4: intermittent dryout dominant region [170]. Reprinted with permission from Ref. [170], *International Journal of Heat and Mass Transfer*, Copyright © 2018 Elsevier Ltd.

**Figure 22 micromachines-14-01451-f022:**
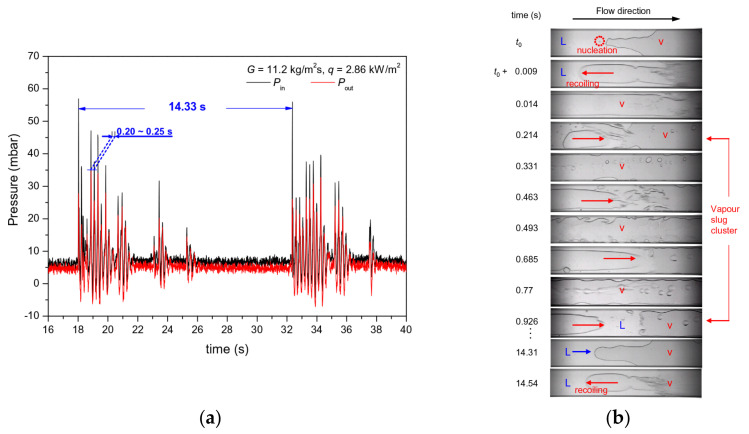
Pressure fluctuation signals and the corresponding flow pattern: (**a**) the distribution of low-frequency and high-amplitude pressure fluctuation signals; (**b**) visualization of the flow boiling [165]. Reprinted with permission from Ref. [165], *International Journal of Heat and Mass Transfer*, Copyright © 2013 Elsevier Ltd.

**Figure 23 micromachines-14-01451-f023:**
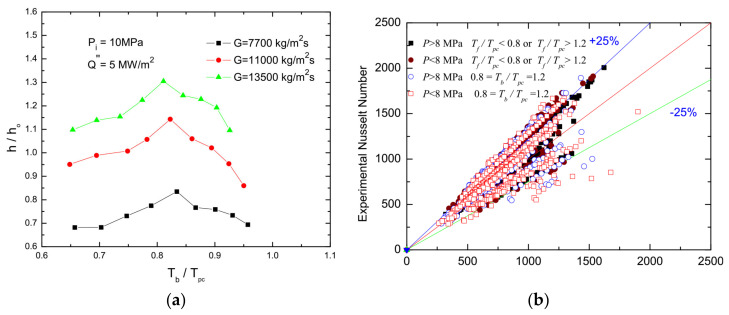
The results of heat transfer enhancement: (**a**) heat transfer enhancement under different mass flow rates; (**b**) new semi-empirical correlation based on PDF method [180]. Reprinted with permission from Ref. [180], *Applied Thermal Engineering*, Copyright © 2013 Elsevier Ltd.

**Figure 24 micromachines-14-01451-f024:**
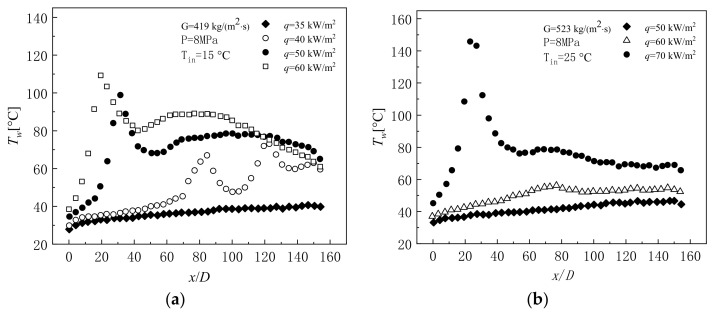
Axial distribution of wall temperature: (**a**) under different mass flow rates; (**b**) under different heat flux [192]. Reprinted with permission from Ref. [192], *International Journal of Heat and Mass Transfer*, Copyright © 2007 Elsevier Ltd.

**Figure 25 micromachines-14-01451-f025:**
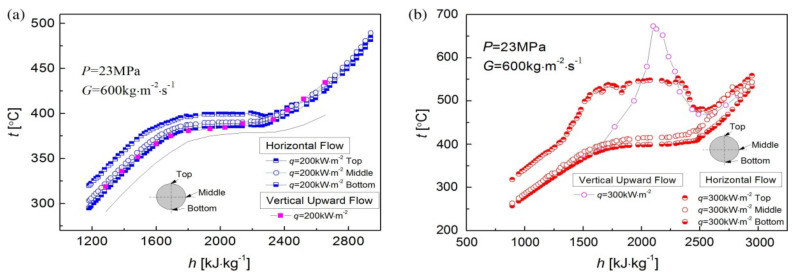
A comparison of inner wall temperature distribution in horizontal and vertical upward flow EHT and DHT modes. (**a**) Enhanced heat transfer ($q/G=0.33\;\mathrm{kJ}\cdot {\mathrm{kg}}^{-1},\;P=23\;\mathrm{MPa}$); (**b**) deteriorated heat transfer ($q/G=0.33\;\mathrm{kJ}\cdot {\mathrm{kg}}^{-1},\;P=23\;\mathrm{MPa}$) [181]. Reprinted with permission from Ref. [181], *Applied Thermal Engineering*, Copyright © 2016 Elsevier Ltd.

**Figure 26 micromachines-14-01451-f026:**
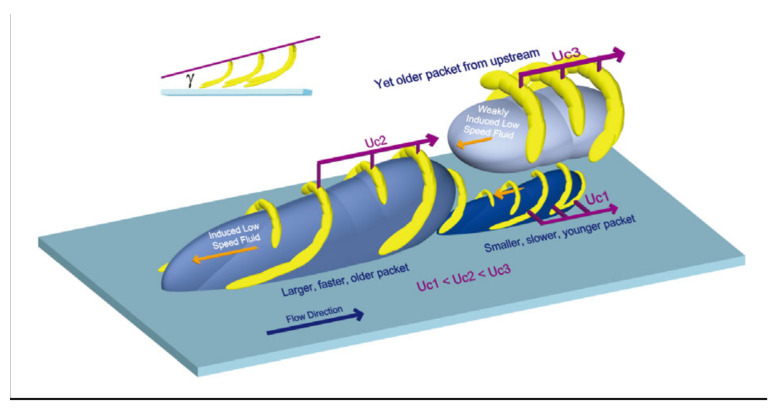
Basic models of large-scale turbulent structures [201]. Article published with open access.

**Figure 27 micromachines-14-01451-f027:**
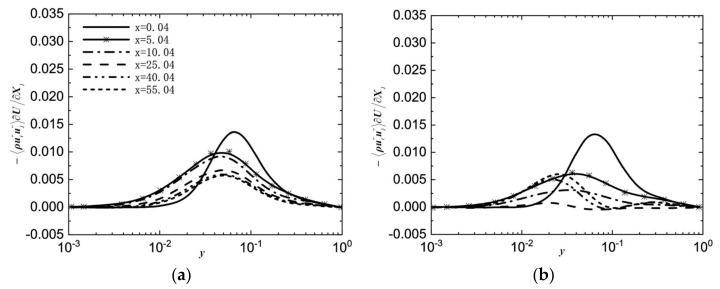
Distribution of kinetic production rate: (**a**) the forced convection; (**b**) the mixed convection [204]. Reprinted with permission from Ref. [204], *Physics of Fluids*, Copyright © 2005 Published by AIP Publishing.

**Figure 28 micromachines-14-01451-f028:**
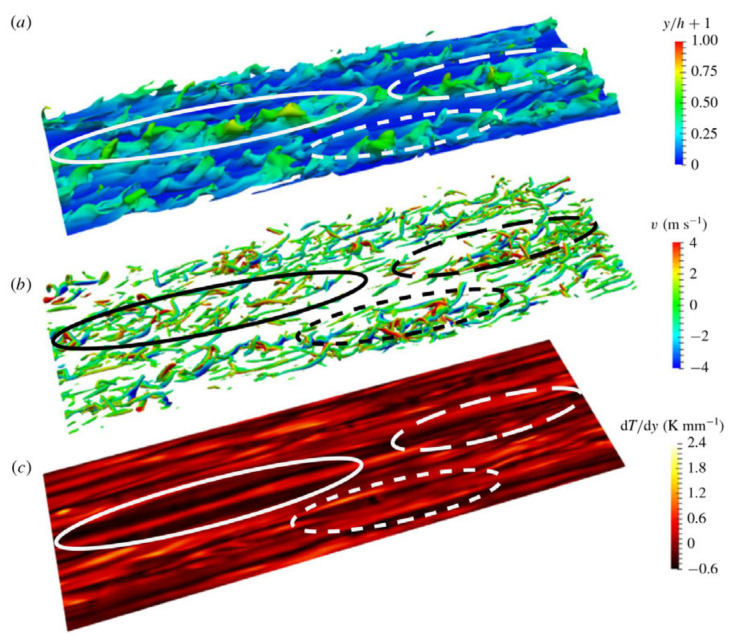
Iso-surfaces of density colored by the distance from the bottom wall (color online): (**a**) Q-criterion colored by the wall-normal velocity; (**b**) temperature gradient; (**c**) $p_{b}=1.1p_{\mathrm{cr}}$ and ΔT=5K [215]. Reprinted with permission from Ref. [215], *The Journal of Fluid Mechanics*, Copyright © 2019 Cambridge University Press.

**Figure 29 micromachines-14-01451-f029:**
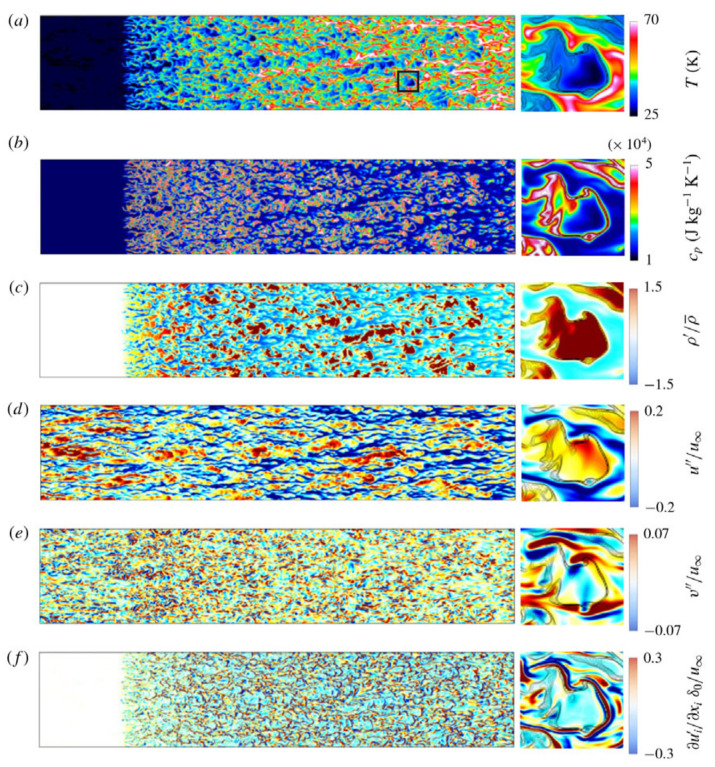
Instantaneous wall-parallel snapshots of the heated transcritical turbulent boundary layer. Schemes follow the same formatting: (**a**) temperature; (**b**) specific heat at constant pressure; (**c**) density fluctuations; (**d**) Favre-averaged streamwise velocity fluctuations; and (**e**) divergence of velocity fluctuations, (**f**) divergence of velocity fluctuations [214]. Reprinted with permission from Ref. [214], *The Journal of Fluid Mechanics*, Copyright © 2019 Cambridge University Press.

**Figure 30 micromachines-14-01451-f030:**
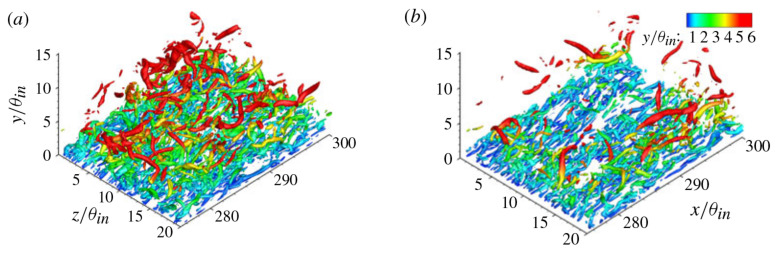
The instantaneous three-dimensional vortical structures: (**a**) unheated wall; (**b**) strongly heated wall [220]. Reprinted with permission from Ref. [220], *The Journal of Fluid Mechanics*, Copyright © 2013 Cambridge University Press.

**Figure 31 micromachines-14-01451-f031:**
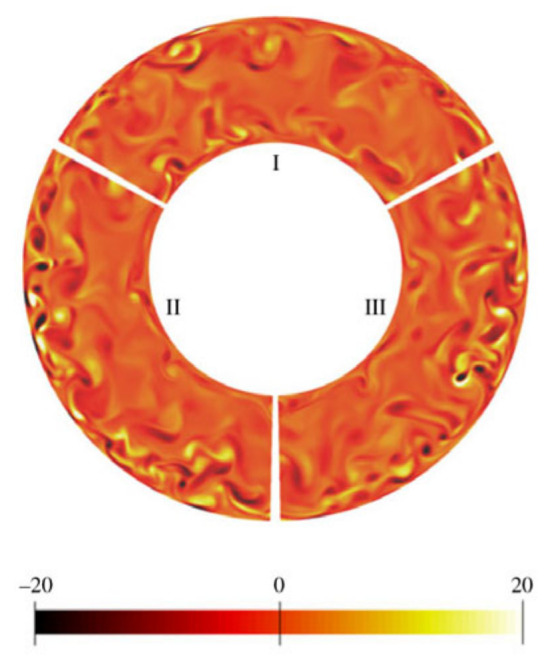
Visualization of flow vortices at the cold outer wall and hot inner wall [223]. Reprinted with permission from Ref. [223], *The Journal of Fluid Mechanics*, Copyright © 2016 Cambridge University Press.

**Figure 32 micromachines-14-01451-f032:**
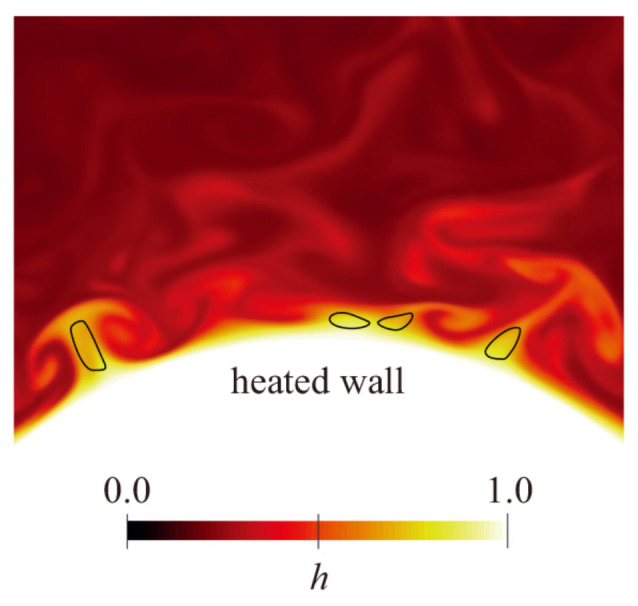
Typical mushroom structures near the hot wall [222]. Reprinted with permission from Ref. [222], *Physical Review Fluids*, Copyrights 2022–2023 Aptara Inc.

**Figure 33 micromachines-14-01451-f033:**
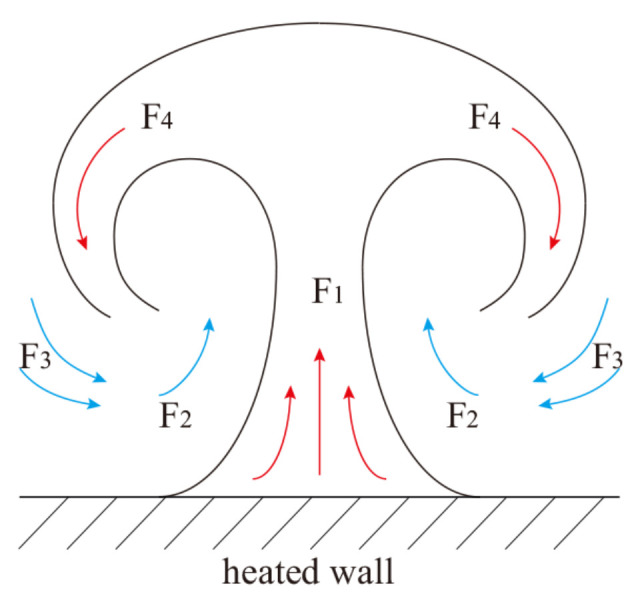
A physical interpretation of the flux quadrants using the mushroom structure [222]. Reprinted with permission from Ref. [222], *Physical Review Fluids*, Copyrights 2022–2023 Aptara Inc.

**Figure 34 micromachines-14-01451-f034:**
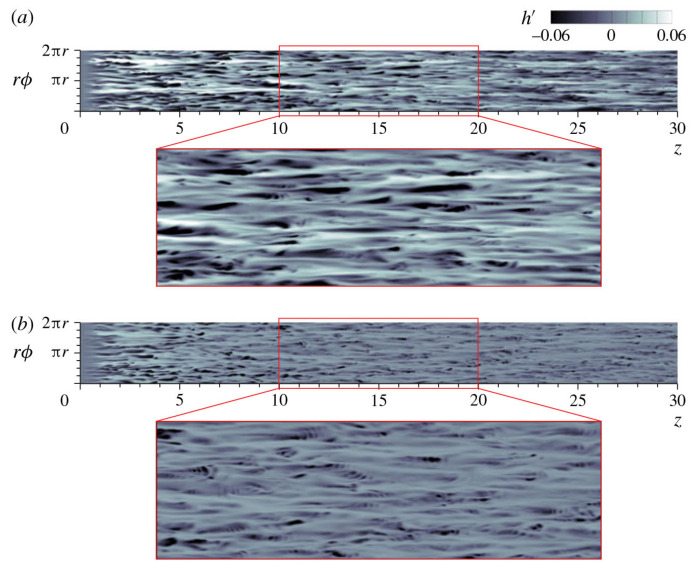
Instantaneous enthalpy fluctuations: (**a**) isoflux condition; (**b**) isothermal condition [209]. Reprinted with permission from Ref. [209], *The Journal of Fluid Mechanics*, Copyright © 2016 Cambridge University Press.

**Figure 35 micromachines-14-01451-f035:**
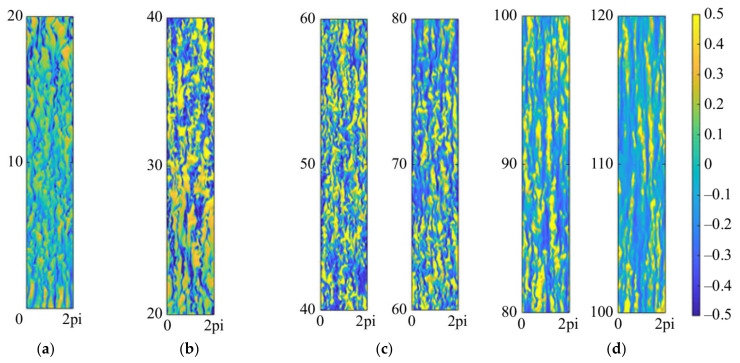
Instantaneous iso-surfaces of the streaks $(\rho u_{x}{)}^{\prime }$ at $y^{+}=20$: (**a**) Period Ⅰ; (**b**) Period Ⅱ; (**c**) Period Ⅲ; and (**d**) Period Ⅳ [228]. Reprinted with permission from Ref. [228], *The Journal of Fluid Mechanics*, Copyright © 2021 Cambridge University Press.

**Figure 36 micromachines-14-01451-f036:**
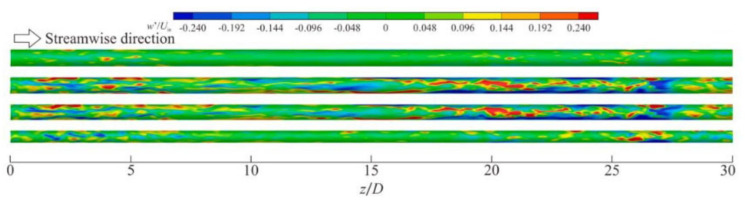
Turbulent streaks along the tube at different non-dimensional wall distances, from top to bottom $y_{\mathrm{ref}}^{+}=3,\;15,\;30\;\mathrm{and}\;100$ [231]. Reprinted with permission from Ref. [231], *International Journal of Heat and Fluid Flow*, Copyright © 2022 Elsevier Inc.

**Figure 37 micromachines-14-01451-f037:**
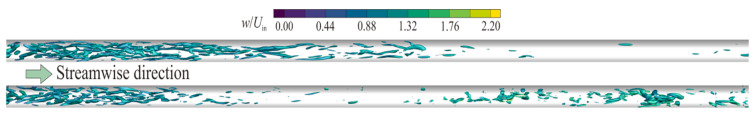
Instantaneous vortex structures by Q-criterion and colored by streamwise velocity [231]. Reprinted with permission from Ref. [231], *International Journal of Heat and Fluid Flow*, Copyright © 2022 Elsevier Inc.

**Table 1 micromachines-14-01451-t001:** Classification criteria for microchannels.

Kandlikar and Grande [28]	
Transitional channel	$0.1\;\mu m<d_{h}\leq 10\;\mu m$
Microchannel	$10\;\mu m<d_{h}\leq 200\;\mu m$
Mini-channel	$200\;\mu m<d_{h}\leq 3\;\mathrm{mm}$
Conventional channel	$3\;\mathrm{mm}<d_{h}$

**Table 2 micromachines-14-01451-t002:** Relationship between force and scale [32].

Type of Force	Scale of Force	Type of Force	Scale of Force
Electromagnetic force	~$L^{4}$	Inertia force	~$L^{2}$
Centrifugal force	~$L^{4}$	Viscous force	~$L^{1}$
Gravity	~$L^{3}$	Surface tension	~$L^{1}$
Buoyancy force	~$L^{3}$	Electrostatic force	~$L^{-2}$

L: characteristic scale.

**Table 3 micromachines-14-01451-t003:** Scale effects in microchannels.

References	Channel Type	Characteristic Scale	Fluids	Flow Pattern	Scale Effects	Remark
Aydin [35]	Channels	$0.001\leq K_{n}\leq 0.1$	Gas	Laminar	Rarefaction	The increase in Knudsen number would lead to a decrease in Nussel number
Tang [36]	Channels	50–300 μm	Nitrogen/Helium	Laminar	Rarefaction/Compressibility/Surface roughness	The comprehensive effects of rarefaction, roughness, and compressibility on the flow characteristics were discussed.
Cetin [37]	Channels	$0\leq K_{n}\leq 0.1$	Constant fluid	Laminar	Rarefaction/Viscous heating	The rarefaction effect was achieved through velocity slip and temperature jump boundary conditions.
Mala [38]	Tubes	50–254 μm	Water	Laminar	Surface roughness	Significant deviations of flow characteristics from the predictions of conventional theory
Hegab [39]	Channels	112–210 μm	R-134a	Transition/turbulent	Surface roughness	Friction loss and heat transfer rate were lower than commonly predicted values using macroscopic scale correlations
Lorenzini [40]	Tubes	26–508 μm	Nitrogen	Laminar/transition/turbulent	Surface roughness	Verified the applicability of flow characteristics under different flow regimes to traditional theories
Sadaghiani [41]	Channel with pins	40 μm	Gas	Laminar	Surface roughness	The recirculation flow generated between roughness units reduced the Nu number and increased the friction coefficient
Mandev [42]	Channels	300–700 μm	Water	Laminar (Re = 10–80)	Surface roughness	The increased heat transfer surface area, mixing effect, and boundary layer interaction were considered to be the reasons for the increased heat transfer due to surface roughness
Lin [43]	Channels	100–600 μm	Air/water	Laminar	Axial heat transfer	Proposed new parameters to evaluate the impact of axial heat conduction effects in walls
Huang [44]	Channels	102 μm	Water	Laminar (Re = 15–80)	Axial heat transfer	Axial heat conduction directed some of the heat towards the entrance and reduces convective heat transfer in the fully developed region
Hsieh [45]	Circle/square/triangle/ellipse/hexagonal tubes	80 μm	Water	Laminar	Axial heat transfer/surface roughness	Axial wall heat conduction was noted for both smooth and roughened channels
Chung [46]	Circular tubes	100 μm	Water/nitrogen	Laminar	Compressibility effects	After considering the compressibility effect, the measured friction coefficient was consistent with the theory.
Vijayalakshmi [47]	Trapezoidal channels	60.5–211 μm	Nitrogen	Laminar/turbulent	Compressibility effects	After proper consideration of the significant effect of compressibility, no special microchannel effect was found, including the early transition to turbulence
Tso [48]	Channels	0.729/0.727 mm	Water	Laminar	Viscous heating	The viscosity variations couple the temperature and velocity fields, causing a change in the temperature profile along and across the flow, which affected the convection
Koo [49]	Pipes/channels	<50 μm	Water, methanol and isopropanol	Laminar	Viscous heating	Viscous dissipation was a strong function of the channel aspect ratio, Reynolds number, Eckert number, Prandtl number, and conduit hydraulic diameter
Cavazzuti [50]	Pipes/channels	100 μm	constant fluid	Laminar	Viscous heating	Established a heat transfer correlation considering viscous effects

**Table 4 micromachines-14-01451-t004:** Classification of flow regimes according to Kn.

*Kn* Number	Regime	Fluid Model
$K_{n}<10^{-3}$	Continuum	Navier–Stokes equations with no slip boundary conditions
$10^{-3}<K_{n}<10^{-1}$	Slip flow	Navier–Stokes equations with velocity slip and temperature jump boundary conditions
$10^{-1}<K_{n}<10$	Transition	Higher order continuum transport equations
$K_{n}>10$	Free molecular flow	Collisionless Boltzmann equation

**Table 5 micromachines-14-01451-t005:** Selected studies on flow boiling heat transfer and mechanisms in microchannels.

References	Channel Type	Fluids	Parameters	Flow Pattens	Remark
Chen [136]	Circular, D_h_ = 1.1–4.26 mm	R134a	P = 6–14 bar T = 20–55 °C g = 0.5–25 kg/h Q = 2.68–1640 W	Dispersed bubble, bubbly, confined bubble, slug, churn, annular and mist flow	The transition boundaries of slugchurn and churn- annular depend strongly on diameter. On the contrary, the dispersed bubble to churn and bubbly to slug boundaries are less affected.
Magnini [137]	Horizontal, circular D_h_ = 0.3–0.7 mm	R245fa	T_sat_ = 10–50 °C G = 400–700 kg/(m^2^s) q_w_ = 5–20 kW/m^2^	slug flow boiling	The peculiar flow structure of a slug flow strongly influences the local and average heat transfer performance via the liquid film thickness, bubble length, and velocity.
Liu [138]	Vertical, rectangular D_h_ = 2.76 mm	R600a, R227ea, R245fa	P_sat_ = 0.162–0.567 MPa T_sat_ = 27.5/36.6/45.5 G = 32.2–116.8 kg/(m^2^s) q_w_ = 5–20 kW/m^2^	Bubble, slug, churn-annular, annular and part- ial dryout-annular flow	Five flow types are identified. New experimental results for heat transfer coefficients, tracking the effects of heat flux, mass flux, vapor quality, and saturation pressure, were presented.
Zhang [139]	Horizontal, circular, D_h_ = 2.92 mm	Nitrogen	P_in_ = 192–350 kPa G = 170–310 kg/(m^2^s) q_w_ = 1.4–43.7 kW/m^2^	nucleate boiling	The heat flux increased nearly exponentially with the wall superheat, and the heat transfer coefficient was independent of mass flux and vapor quality but dependent on heat flux and working pressure.
Lv [140]	Horizontal micro-tube circular	R134a Ethane	P = 0.3–1.1 MPa T_in_ = 190–230 K q_w_ = 5–60 kW/m^2^	bubble flow and slug flow	The law of boiling heat transfer characteristics with vapor quality, heat flux, mass flow, ethane mole fraction, and other factors are obtained.
Harirchian [141]	Parallel microchannels 100–5850 μm	FC-77	P = 0.3–1.1 MPa T_in_ = 190–230 K q_w_ = 0–380 kW/m^2^	bubbly, slug, churn, wispy-annular, annular flows	Flow regimes in the smaller microchannels are different, and bubble nucleation at the walls is suppressed at a relatively low heat flux for these sizes.
Criscuolo [142]	Multi-microchannels 298×1176 μm	R1234yf R1234ze(E) R134a	T_sat_ = 30.5–40.5 °C G = 415–1153 kg/(m^2^s) q_w_ = 3–145 W/cm^2^	bubbly flow, churn flow, wispy-annular flow, churn flow	Three boiling regimes (I, II, III) were identified according to the observed effect of the heat and mass flux.
Saisorn [143]	Horizontal/vertical, circular D_h_ = 1 mm	R134a	P_sat_ = 8.0 MPa G = 250–820 kg/(m^2^s) q_w_ = 4–60 kW/m^2^	Slug flow, throat- annular flow, churn flow, annular flow, annular-rivulet flow	The experimental results showed the importance of the change in the flow direction. The shape of the gas slug during horizontal flow did not look the same as in the vertical orientations.

**Table 6 micromachines-14-01451-t006:** Research on heat transfer enhancement and heat transfer deterioration.

References	Channel Type	Diameter	Fluids	Main Parameters	Heat Transfer Pattens
Dang [185]	Horizontal, circular tubes	D_h_ = 1–2 mm	CO_2_	P = 8–10 MPa T_in_ = 30–70 °C G = 800–1200 kg/(m^2^s) q_w_ = 12 kW/m^2^	Enhancement
Wang [179]	Horizontal, circular tubes	D_h_ = 0.5–1 mm	CO_2_	P = 76.6–90 bar T_in_ = 30.9–37.3 °C G = 672–4810 kg/(m^2^s) q_w_ = 21–128.7 kW/m^2^	Enhancement
Gu [180]	Horizontal, circular tubes	D_h_ = 1.6 mm	Methane	P = 5–15 MPa T_in_ = -150 °C G = 60–150 × 10^3^ kg/(m^2^s) q_w_ = 1–16 MW/m^2^	Enhancement
Li [186]	Horizontal, circular tubes	D_h_ = 0.8 mm	Methane	P = 4.77–7.31 MPa T_in_ = 170 K g = 0.124–0.198 g/s q_w_ = 22–35 kW/m^2^	Enhancement
Lei [181]	Vertical/horizontal, circular tubes	D_h_ = 3 mm	Water	P = 23–28 MPa G = 200–600 kg/(m^2^s) q_w_ = 0–400 kW/m^2^	Deterioration
Peng [182]	Vertical, circular tubes	D_h_ = 2 mm	CO_2_	P = 7.5–9 MPa T_in_ = 7–9 °C G = 100–1200 kg/(m^2^s) q_w_ = 45–300 kW/m^2^	Deterioration`
Liu [183]	Vertical, circular tubes	D_h_ = 0.9/2 mm	n-decane	P = 3–5 MPa T_in_ = 150 °C G = 341–7378 kg/(m^2^s) q_w_ = 53–406 kW/m^2^	Deterioration`
Zhu [184]	Vertical, square tubes	D_h_ = 1.8 mm	n-decane	P = 3–5 MPa T_in_ = 473 K g = 1.0–2.0 g/s q_w_ = 100–500 kW/m^2^	Deterioration

**Table 7 micromachines-14-01451-t007:** Summary of the heat transfer correlations in microchannels.

Reference	Flow Direction	Correlations	Fluid and Parameters
Bishop [238]	-	Nub=0.0069Reb0.9Prb0.66(ρwρb)0.43[1+2.4d/L]	Water D = 2.5–4.1 mm
Liao [189]	Horizontal	Nub=0.124Reb0.8Prb0.4GrReb20.203ρwρb0.842c¯pcp,b0.384	CO_2_ D = 0.7–2.16 mm P = 7.4–12 MPa T_in_ = 20–110 °C $\dot{m}\;$ = 200–1200 kg/m^2^ s
	Upward	Nub=0.354Reb0.8Prb0.4GrmReb2.70.157ρwρb1.297c¯pcp,b0.296	
	Downward	Nub=0.643Reb0.8Prb0.4GrmReb2.70.186ρwρb2.154c¯pcp,b0.751	
Dang [185]	Horizontal	Nu=ff/8Reb−1000Pr1.07+12.7ff/8Pr2/3−1 $\mathrm{Pr}=\left\{\begin{array}{l} c_{p_{b}}\mu_{b}/k_{b},\;\mathrm{for}\;c_{p_{b}}\geq {\bar{c}}_{p}\\ {\bar{c}}_{p}\mu_{b}/k_{b},\;\mathrm{for}\;c_{p_{b}}<{\bar{c}}_{p}\;\mathrm{and}\;\mu_{b}/k_{b}\geq \mu_{f}/k_{f}\\ {\bar{c}}_{p}\mu_{f}/k_{f},\;\mathrm{for}\;c_{p_{b}}<{\bar{c}}_{p}\;\mathrm{and}\;\mu_{b}/k_{b}<\mu_{f}/k_{f} \end{array}\right.$ c¯p=hb−hw/Tb−TwReb=Gd/μbff=1.82log10Ref−1.64−2Ref=Gd/μf	CO_2_ D = 1/2/4/6 mm P = 8/9/10 MPa T_in_ = 30–70 °C $\dot{m}\;$ = 200/−1200 kg/m^2^ s q_w_ = 6–33 kW/m^2^
Wang [179]	Horizontal	Nuz=0.225Ref,z0.423Prf,z0.229Bof,z*−0.156T*0.055C¯pzCpf,z0.401 Bof,z∗=Grf,zRef,z2, $T^{*}=\frac{T_{\mathrm{pc}}-T_{f,\;\mathrm{in}\;}}{T_{f,\;\mathrm{out}\;}-T_{f,\;\mathrm{in}\;}}$ ${\mathrm{Gr}}_{f,z}=\frac{\left(\rho_{w,z}-\rho_{f,z}\right)g\rho_{f,z}D_{s,i}^{3}}{\mu_{f,z}^{2}}$	CO_2_ D = 0.5/0.75/1.0 mm P = 7.66–9.0 MPa T_in_ = 30.8–37.3 °C $\dot{m}\;$ = 672–4810 kg/m^2^ s q_w_ = 70.7–344.2 kW/m^2^
Peng [182]	Upward	Nu=0.19811Reb0.98476Prb¯1.05313ρwρb0.59422Bu0.1q+0.45786 Bu=Grb¯Reb2.7, $\bar{{\mathrm{Gr}}_{b}}=\frac{\left(\rho_{b}-\bar{\rho }\right)\rho_{b}{\mathrm{gd}}^{3}}{\mu_{b}^{2}}$	CO_2_ D = 2.0 mm P = 7.5–9.0 MPa T_in_ = 30.8–37.3 °C $\dot{m}\;$ = 100–1200 kg/m^2^ s q_w_ = 45–300 kW/m^2^
Gu [180]	Horizontal	Nub,pdf=0.023Reb,pdf0.8Prb,pdf0.4(ρwρb,pdf)0.3(Cppdf¯Cpb,pdf)n	Methane D = 1.6 mm P = 5–15 MPa T_in_ = 30.8–37.3 °C $\dot{m}\;$ = 6–150 $\times 10^{3}$ kg/m^2^ s q_w_ = 1–16 MW/m^2^
Liu [183]	Upward/downward	$\frac{\mathrm{Nu}}{{\mathrm{Nu}}_{f}}={\left[\left|1+A\cdot {\mathrm{Bo}}^{*}{\left(\frac{{\bar{c}}_{p}}{c_{\mathrm{pb}}}\right)}^{a}{\left(\frac{\rho }{\rho_{b}}\right)}^{b}{\left(\frac{\mathrm{Nu}}{{\mathrm{Nu}}_{f}}\right)}^{-2}\right|\right]}^{0.46}$ Upward: $A=-8.45\times 10^{5},\;a=-2.23,\;b=0.14$ Downward: $A=3.62\times 10^{5},a=-2.75,\;b=0.14$	N-decane D = 0.95/2 mm P = 2.5–7 MPa T_in_ = 16–500 °C
Fu [237]	Upward/downward	Upward: $\frac{\mathrm{Nu}}{{\mathrm{Nu}}_{f}}=0.18{\left(10^{7}\cdot {\mathrm{Bo}}^{*}\right)}^{0.1}{\left(\frac{\rho_{w}}{\rho_{b}}\right)}^{0.08}{\left(\frac{\eta_{w}}{\eta_{b}}\right)}^{0.1}\left[1+6.8{\left(\frac{\mathrm{Nu}}{{\mathrm{Nu}}_{f}}\right)}^{1.12}\right]$ Downward: $\frac{\mathrm{Nu}}{{\mathrm{Nu}}_{f}}=0.178{\left(10^{7}\cdot {\mathrm{Bo}}^{*}\right)}^{0.1}{\left(\frac{\rho_{w}}{\rho_{b}}\right)}^{0.11}{\left(\frac{\eta_{w}}{\eta_{b}}\right)}^{0.12}\left[1+7.12{\left(\frac{\mathrm{Nu}}{{\mathrm{Nu}}_{f}}\right)}^{1.09}\right]$	RP-3 D = 1.09 mm $\dot{m}\;$ = 1.0–2.0 g/s q_w_ = 90.5–600 kW/m^2^

## Data Availability

All data are taken from the literature and are available in the referenced papers.

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
