# Peer review of "A Review of the Complex Flow and Heat Transfer Characteristics in Microchannels"

_micromachines, 2023, doi:10.3390/mi14071451_

Round 1
Reviewer 1 Report
1. Lines 101-134: The authors discussed the impact of the "scale effect" on the flow and heat transfer performance of microchannels. The first paragraph of section 2.2 first mentions the "scale effect", while the second paragraph discusses what is "scale effect" is and the physical mechanism of the "scale effect". Is the logic of the discussion in the two paragraphs reasonable?
2. Lines 138-150: Section 2.2.1 refers to the "rarefied effect", but the authors did not give a basic explanation of the "rarefied effect".
3. Lines 319-345: FIG. 10 presents the effects of surface roughness at different heights on flow and heat transfer characteristics under different Reynolds numbers, both of which are positively correlated with roughness heights. However, the correlation and relationship shown by the curve are different from the above conclusions.
4. Lines 503-681: Section 3 is titled "Mechanism of flow heat transfer under the coupling effect of scale effect and special fluid", but Section 3.1 spends a lot of space on boiling heat transfer, and then talks about the coupling effect of scale effect and special fluid. Is the structure of this part of the article reasonable?
5. Lines 714-764: Section 3.2.1 entitled "Research on Enhanced Heat Transfer Characteristics of Supercritical Pressure Fluids in Microchannels"; and Section 3.2.2 entitled "Study of heat transfer degradation characteristics of Supercritical pressure fluids in Microchannels", both investigated the heat transfer characteristics of supercritical pressure fluids in microchannels. In fact, section 3.2.2 only discussed the deterioration characteristics.
6. Lines 899-932: FIG. 29 shows the downflow velocity fluctuations and wall-normal velocity fluctuations of the turbulent jet (Q2) and turbulent swept flow (Q4). However, the authors did not elaborate on the meaning and existence significance of the module (Q1) in the figure.
Need to be corrected carefully.
Reviewer 2 Report
The reviewers have presented details of a great number of studies on a variety of different subtopics related to the review title.
However, for a review to be successful, one needs to summarize and synthesize the results so that the reader can easily understand the main aspects of the earlier studies.
This was not done in this study, apart from the last section, which was a nice summary, but there were no actual discussion of what was done in the reviewed works.
I think the paper can be remedied, but only after:
- extensive addition of text which summarizes each section, with, for example, tables (as was done in the last section), or comparion graphs between different studies; and,
- in the last section, a description of the studies which derived these correlations needs to be added, including the sources of data used to create them - that is, were the data collected as part of the study, was it a summary of data from the open literature, or a combination of both?
Reads well, minor typos.
Reviewer 3 Report
1. Table 2, please explain the meaning of L^* by note below table 2.
2. Please make up the definition of the rarefaction effects.
3. If some figures of this work were from literatures, please mark the cited literatures in the title of figures.
4. In the section of the liquid with phase change in microchannels, if the review on condensation heat transfer can be added, it would be great.
The english expression is great.
Round 2
Reviewer 1 Report
This article can be published.
Author Response
Thanks for your careful work.
Reviewer 2 Report
It appears that the authors have added some summaries in the form of tables in the paper, but they have not summarized them in the text. Also, the final section is still not discussed in any detail.
Reads well, minor typos.
Author Response
Thanks for your careful work. Please see the attachment.

Round 3
Reviewer 2 Report
Changes acceptable